# AlphaEdit: Null-Space Constrained Knowledge Editing for Language Models

**Junfeng Fang**[1][*]**, Houcheng Jiang**[1][*]**, Kun Wang**[1]**, Yunshan Ma**[2]**,**
**Jie Shi**[2]**, Xiang Wang**[1][†]**, Xiangnan He**[1][†]**, Tat-Seng Chua**[2]
[1]University of Science and Technology of China, [2]National University of Singapore
`fangjf1997@gmail.com`, `jianghc@mail.ustc.edu.cn`

## ABSTRACT

Large language models (LLMs) often exhibit hallucinations, producing incorrect or outdated knowledge. Hence, model editing methods have emerged to enable targeted knowledge updates. To achieve this, a prevailing paradigm is the locating-then-editing approach, which first locates influential parameters and then edits them by introducing a perturbation. While effective, current studies have demonstrated that this perturbation inevitably disrupts the originally preserved knowledge within LLMs, especially in sequential editing scenarios. To address this, we introduce AlphaEdit, a novel solution that projects perturbation into the null space of the preserved knowledge before applying it to the parameters. We theoretically prove that this projection ensures the output of post-edited LLMs remains unchanged when querying about the preserved knowledge, thereby mitigating the issue of disruption. Extensive experiments on various LLMs, including LLaMA3, GPT2-XL, and GPT-J, show that AlphaEdit boosts the performance of most locating-then-editing methods by an average of $36.7\%$ with **a single line of additional code for projection solely**. Our code is available at: https://github.com/jianghoucheng/AlphaEdit.

## 1 INTRODUCTION

Large language models (LLMs) have demonstrated strong capability to store extensive knowledge during pre-training and recall it during inference (Brown et al., 2020; Petroni et al., 2019; Roberts et al., 2020; Liu et al., 2024a). Despite this, they frequently exhibit hallucinations, producing incorrect or outdated information (Cao et al., 2021; Mitchell et al., 2022a). While fine-tuning with updated knowledge offers a straightforward solution, it is often prohibitively time-consuming, especially for resource-limited scenarios. In sight of this, model editing methods have emerged, enabling updating the target knowledge while preserving other knowledge (Meng et al., 2022; Yao et al., 2023). Broadly, model editing approaches fall into two categories: (1) parameter-modifying methods, which directly adjust a small subset of parameters (Meng et al., 2023; Jiang et al., 2025; Li et al., 2024b), and (2) parameter-preserving methods that integrate additional modules without altering the original parameters (Huang et al., 2023; Yu et al., 2024; Hartvigsen et al., 2023; Zheng et al., 2023).

In this paper, we explore the parameter-modifying methods for model editing which require no additional memory. Concretely, current parameter-modifying methods typically follow the locate-then-edit paradigm (Meng et al., 2022). The basic idea is to first locate influential parameters $W$ through causal tracing, and then edit them by introducing a perturbation $\Delta$. The common objective for solving $\Delta$ is to minimize the output error on the to-be-updated knowledge, denoted as $e_1$. Additionally, the output error on the to-be-preserved knowledge, $e_0$, also needs to be considered in the objective function to ensure the model behavior unchanged on the preserved knowledge.

Despite their success, the current paradigm faces a critical limitation: it struggles to maintain a balance between knowledge-update error $e_1$ and knowledge-preservation error $e_0$. Specifically, to prioritize the success of update, prior studies focus more on minimizing $e_1$ by assigning a larger weight, while taking insufficient control over $e_0$. This could make the LLM after editing (*i.e.,* the

---

[*]Equal contribution. [†]Corresponding author: {`xiangwang1223, xiangnanhe`}@gmail.com.

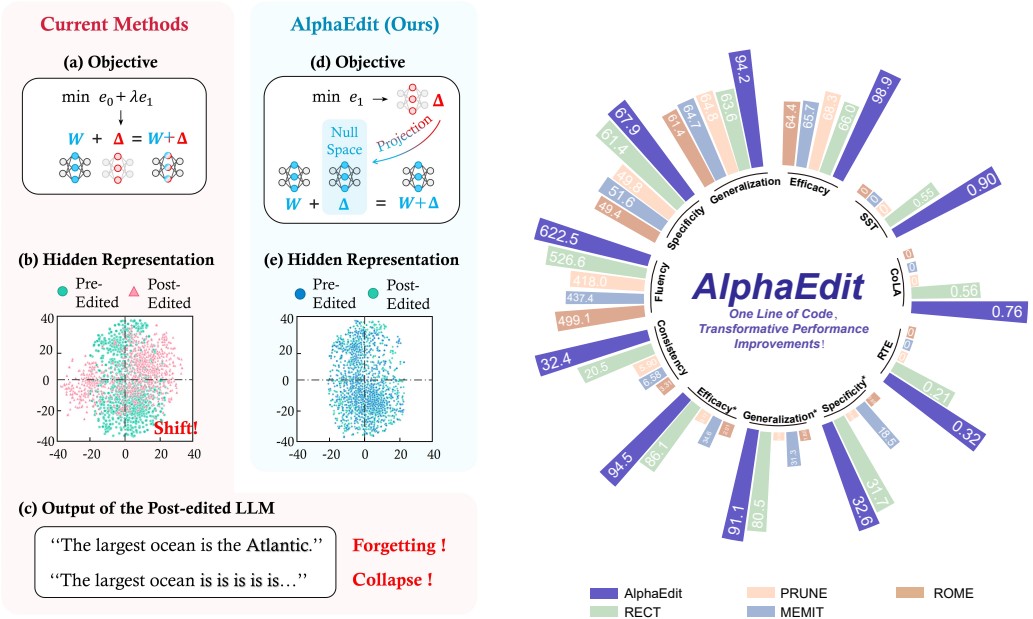

Figure 1: Comparison between the current methods and AlphaEdit. (a) and (d) exhibit the objectives, where $\lambda$ is the coefficient to keep balance between $e_0$ and $e_1$ in the objective; (b) and (e) show the distributions of hidden representations after dimensionality reduction within the pre-edited and post-edited LLaMA3, respectively; (c) depicts the output of the post-edited LLaMA3. Best viewed in color.

Figure 2: Performance of various model editing methods on LLaMA3 (8B). Results with asterisks in the superscript are from the ZsRE dataset. SST, RTE, and CoLA demonstrate the general capabilities of the post-edited LLMs. The experiments are conducted with 2000 edited samples for sequential editing. Detailed settings and results are provided in Section 4.2 and Table 1, respectively. Best viewed in color.

post-edited LLM) overfit to the updated knowledge. This overfitting would introduce a distribution shift of the hidden representations within LLMs. Figure 1 (b) showcases this shift, where the hidden representations in the post-edited LLaMA-3 (8B) (Meta, 2024) diverge from the original distribution (*i.e.,* the pre-edited LLM). Worse still, in sequential editing scenarios (Gupta & Anumanchipalli, 2024) where the LLM undergoes multiple sequential edits, the accumulation of overfitting destroys the model's ability to preserve knowledge and generate coherent sentences, eventually leading to model forgetting and model collapse, as depicted in Figure 1 (c).

To address these flaws, we instead remove $e_0$ from the current objective, allowing the model to focus solely on minimizing $e_1$ without trade-offs. To avoid overfitting to the to-be-updated knowledge, we project the solution of this new objective into the **null space** (Wang et al., 2021) of the preserved knowledge before applying it to the model parameters, as shown in Figure 1 (d). By leveraging the mathematical properties of matrix projection and null space, our new objective ensures that the distribution of hidden representations within LLMs remains invariant after editing, as shown in Figure 1 (e). This invariance enables the post-edited LLM to reduce $e_1$ while keeping $e_0$ close to zero, thereby alleviating the issues of model forgetting and model collapse. We theoretically prove such invariance property in Section 3. In a nutshell, we term the method as **AlphaEdit**, a simple yet effective editing approach with a null-space constraint for LLMs.

To validate the effectiveness of our method, we conducted extensive experiments on multiple representative LLMs, such as GPT-2 XL (Radford et al., 2019) and LLaMA-3 (8B). The results show that, compared to the best-performing baseline, **AlphaEdit can achieve an average performance improvement of 36.7% by adding just one line of code** to the conventional model editing method, MEMIT (Meng et al., 2023), as illustrated in Figure 2. Furthermore, we empirically verified that this simple idea can be easily applied to most existing model editing methods (Meng et al., 2022; 2023; Ma et al., 2025; Gu et al., 2024; Li et al., 2024b), functioning as a plug-and-play enhancement that significantly boosts their performance. This highlights AlphaEdit's crucial role in efficient knowledge updates for LLMs, enabling broader applications and future advancements in the field.

## 2 PRELIMINARY

### 2.1 AUTOREGRESSIVE LANGUAGE MODEL

An autoregressive LLM predicts the next token $x$ in a sequence based on the preceding tokens. Specifically, the hidden state of $x$ at layer $l$ within the model, denoted as $h^l$, can be calculated as:

$$h^l = h^{l-1} + a^l + m^l, \quad m^l = W_{\text{out}}^l \, \sigma(W_{\text{in}}^l \, \gamma(h^{l-1} + a^l)), \tag{1}$$

where $a^l$ and $m^l$ represent the outputs of the attention block and the feed-forward network (FFN) layer, respectively; $W_{\text{in}}^l$ and $W_{\text{out}}^l$ are the weight matrices of the FFN layers; $\sigma$ is the non-linear activation function, and $\gamma$ denotes the layer normalization. Following Meng et al. (2022), we express the attention and FFN modules in parallel here.

It is worth noting that $W_{\text{out}}^l$ within FFN layers is often interpreted as a linear associative memory, functioning as key-value storage for information retrieval (Geva et al., 2021). Specifically, if the knowledge stored in LLMs is formalized as $(s, r, o)$ — representing subject $s$, relation $r$, and object $o$ (*e.g.*, $s$ = "The latest Olympic Game", $r$ = "was held in", $o$ = "Paris") — $W_{\text{out}}^l$ associates a set of input keys $k$ encoding $(s, r)$ with corresponding values $v$ encoding $(o)$. That is,

$$\underbrace{m^l}_{v} = W_{\text{out}}^l \underbrace{\sigma(W_{\text{in}}^l \, \gamma(h^{l-1} + a^l))}_{k}. \tag{2}$$

This interpretation has inspired most model editing methods to modify the FFN layers for knowledge updates (Hase et al., 2023; Li et al., 2024a; Hu et al., 2024). For simplicity, we use $W$ to refer to $W_{\text{out}}^l$ in the following sections.

### 2.2 MODEL EDITING IN LLMS

Model editing aims to update the knowledge stored in LLMs through a single edit or multiple sequential edits. Each edit modifies the model parameters $W$ by adding a perturbation $\Delta$ in locate-then-edit paradigm. Specifically, suppose each edit needs to update $u$ pieces of knowledge in the form of $(s, r, o)$. The perturbed $W$ is expected to associate $u$ new $k$-$v$ pairs, where $k$ and $v$ encode $(s, r)$ and $(o)$ of the new knowledge, respectively. Let $W \in \mathbb{R}^{d_1 \times d_0}$, where $d_0$ and $d_1$ represent the dimensions of the FFN's intermediate and output layers. Then, we can stack these keys and values into matrices following:

$$K_1 = [k_1 \,|\, k_2 \,|\, \dots \,|\, k_u] \in \mathbb{R}^{d_0 \times u}, \quad V_1 = [v_1 \,|\, v_2 \,|\, \dots \,|\, v_u] \in \mathbb{R}^{d_1 \times u}, \tag{3}$$

where the subscripts of $k$ and $v$ represent the index of the to-be-updated knowledge. Based on these, the objective can be expressed as:

$$\Delta = \arg\min_{\tilde{\Delta}} \left\| (W + \tilde{\Delta}) K_1 - V_1 \right\|^2, \tag{4}$$

where $\|\cdot\|^2$ denotes the sum of the squared elements in the matrix.

Additionally, let $K_0$ and $V_0$ represent the matrices formed by stacking the $k$ and $v$ of the preserved knowledge. Current methods (Meng et al., 2023; Gu et al., 2024) typically incorporate the error involving $K_0$ and $V_0$ to preserve it, as follows:

$$\Delta = \arg\min_{\tilde{\Delta}} \left( \left\| (W + \tilde{\Delta}) K_1 - V_1 \right\|^2 + \left\| (W + \tilde{\Delta}) K_0 - V_0 \right\|^2 \right). \tag{5}$$

Since $K_0$ and $V_0$ encode the preserved knowledge in LLMs, we have $W K_0 = V_0$ (*cf.* Eqn. 2). Thus, by applying the normal equation (Lang, 2012), if the closed-form solution of Eqn. 5 exists, it can be written as:

$$\Delta = (V_1 - W K_1) K_1^T \left( K_0 K_0^T + K_1 K_1^T \right)^{-1}. \tag{6}$$

Although $K_0$ is difficult to obtain directly since we hardly have access to the LLM's full extent of knowledge, it can be estimated using abundant text input (Meng et al., 2023). In practical applications, $100,000$ $(s, r, o)$ triplets from Wikipedia are typically randomly selected to encode $K_0$ (Meng et al., 2023), making $K_0$ a high-dimensional matrix with $100,000$ columns (*i.e.*, $K_0 \in \mathbb{R}^{d_0 \times 100,000}$). See Appendix B.1 for detailed implementation steps.

## 3 METHOD

In this section, we first introduce the concept of the null space and its relationship to model editing (Section 3.1). Based on this, we present the method for projecting the perturbation $\boldsymbol{\Delta}$ into the null space of the matrix $\boldsymbol{K}_0$, which encodes the persevered knowledge (Section 3.2). Following that, we present AlphaEdit that incorporates the aforementioned projection method in Section 3.3.

### 3.1 NULL SPACE

Null space is at the core of our work. Here we first introduce the definition of the left null space (hereafter referred to simply as *null space*). Specifically, given two matrices $\boldsymbol{A}$ and $\boldsymbol{B}$, $\boldsymbol{B}$ is in the null space of $\boldsymbol{A}$ if and only if $\boldsymbol{BA} = \boldsymbol{0}$. See Adam-NSCL (Wang et al., 2021) for more details.

In the context of model editing, if the perturbation $\boldsymbol{\Delta}$ is projected into the null space of $\boldsymbol{K}_0$ (*i.e.*, $\boldsymbol{\Delta}'\boldsymbol{K}_0 = \boldsymbol{0}$, where $\boldsymbol{\Delta}'$ denotes the projected perturbation), adding it to the parameters $\boldsymbol{W}$ results in:

$$(\boldsymbol{W} + \boldsymbol{\Delta}')\boldsymbol{K}_0 = \boldsymbol{W}\boldsymbol{K}_0 = \boldsymbol{V}_0. \tag{7}$$

This implies that **the projected $\boldsymbol{\Delta}$ will not disrupt the key-value associations of the preserved knowledge** (*i.e.*, $\{\boldsymbol{K}_0, \boldsymbol{V}_0\}$), ensuring the stability of the preserved knowledge.

Therefore, in this work, before adding perturbation $\boldsymbol{\Delta}$ to the model parameters $\boldsymbol{W}$, we project it into the null space of $\boldsymbol{K}_0$ to protect the preserved knowledge. This protection allows us to remove the second term — the term focusing on preserving knowledge — from the objective in Eqn. 5.

### 3.2 NULL SPACE PROJECTING

In Section 3.1, we briefly explained why $\boldsymbol{\Delta}$ should be projected into the null space of $\boldsymbol{K}_0$. In this part, we focus on how to implement this projection.

As introduced at the end of Section 2.2, the matrix $\boldsymbol{K}_0 \in \mathbb{R}^{d_0 \times 100,000}$ has a high dimensionality with $100,000$ columns. Hence, directly projecting the given perturbation $\boldsymbol{\Delta}$ into the null space of $\boldsymbol{K}_0$ presents significant computational challenges. In sight of this, we adopt the null space of the non-central covariance matrix $\boldsymbol{K}_0\boldsymbol{K}_0^T \in \mathbb{R}^{d_0 \times d_0}$ as a substitute to reduce computational complexity, as $d_0$ is typically much smaller than $100,000$. This matrix's null space is equal to that of $\boldsymbol{K}_0$ (please see Appendix B.2 for detailed proof).

Following the existing methods for conducting null space projection (Wang et al., 2021), we first apply a Singular Value Decomposition (SVD) to $\boldsymbol{K}_0\boldsymbol{K}_0^T$:

$$\left\{\boldsymbol{U}, \boldsymbol{\Lambda}, \boldsymbol{U}^T\right\} = \text{SVD}\left(\boldsymbol{K}_0\boldsymbol{K}_0^T\right), \tag{8}$$

where each column in $\boldsymbol{U}$ is an eigenvector of $\boldsymbol{K}_0\boldsymbol{K}_0^T$. Then, we remove the eigenvectors in $\boldsymbol{U}$ that correspond to non-zero eigenvalues[1], and define the remaining submatrix as $\hat{\boldsymbol{U}}$. Based on this, the projection matrix $\boldsymbol{P}$ can be defined as follows:

$$\boldsymbol{P} = \hat{\boldsymbol{U}}\hat{\boldsymbol{U}}^T. \tag{9}$$

This projection matrix can map the column vectors of $\boldsymbol{\Delta}$ into the null space of $\boldsymbol{K}_0\boldsymbol{K}_0^T$, as it satisfies the condition $\boldsymbol{\Delta P} \cdot \boldsymbol{K}_0\boldsymbol{K}_0^T = \boldsymbol{0}$. The detailed derivation is given in Appendix B.3.

Since $\boldsymbol{K}_0$ and $\boldsymbol{K}_0\boldsymbol{K}_0^T$ share the same null space, we can derive $\boldsymbol{\Delta P} \cdot \boldsymbol{K}_0 = \boldsymbol{0}$. Hence, we have:

$$(\boldsymbol{W} + \boldsymbol{\Delta P})\boldsymbol{K}_0 = \boldsymbol{W}\boldsymbol{K}_0 = \boldsymbol{V}_0. \tag{10}$$

This shows that the projection matrix $\boldsymbol{P}$ ensures the model edits not affecting the preserved knowledge in LLMs.

### 3.3 NULL-SPACE CONSTRAINED MODEL EDITING

Section 3.2 has provided how to apply projection to ensure that preserved knowledge is not disrupted. Here, we introduce how to leverage this projection to optimize the current model editing objective.

---

[1]Given that eigenvalues are rarely strictly zero in practical applications, in our experiments, we remove the eigenvectors corresponding to the eigenvalues above $10^{-2}$.

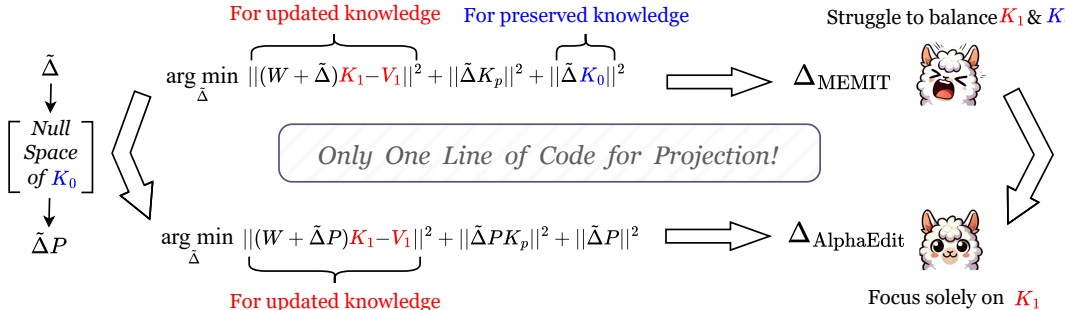

Figure 3: Comparison between the paradigms of AlphaEdit and current method. Best viewed in color.

Starting with the single-edit objective in Eqn. 5, the optimization follows three steps: (1) Replace $\Delta$ with the projected perturbation $\Delta P$, ensuring that the perturbation would not disrupt the preserved knowledge; (2) Remove the first term involving $K_0$, as Step (1) has already protected the preserved knowledge from being disrupted; (3) Add a regularization term $||\Delta P||^2$ to guarantee stable convergence. With these optimizations, Eqn. 5 becomes:

$$\Delta = \arg\min_{\tilde{\Delta}} \left( ||(W + \tilde{\Delta}P)K_1 - V_1||^2 + ||\tilde{\Delta}P||^2 \right), \tag{11}$$

where $K_1$ and $V_1$ denote the key and value matrices of to-be-updated knowledge defined in Eqn. 3.

In sequential editing tasks, during the current edit, we also need to add a term to the objective (*cf.* Eqn. 11) to prevent the perturbation from disrupting the updated knowledge in previous edits. Let $K_p$ and $V_p$ present the key and value matrices of the previously updated knowledge, analogous to the earlier definitions of $K_1$ and $V_1$. This term should minimize $||(W + \tilde{\Delta}P)K_p - V_p||^2$ to protect the association. Since the related knowledge has been updated in previous edits, we have $WK_p = V_p$. Hence, this term can be simplified to $||\tilde{\Delta}PK_p||^2$, and adding it to Eqn. 11 gives the new objective:

$$\Delta = \arg\min_{\tilde{\Delta}} \left( ||(W + \tilde{\Delta}P)K_1 - V_1||^2 + ||\tilde{\Delta}P||^2 + ||\tilde{\Delta}PK_p||^2 \right). \tag{12}$$

To facilitate expression, we define the residual vector of the current edit as $R = V_1 - WK_1$. Based on this, Eqn. 12 can be solved using the normal equation (Lang, 2012):

$$(\Delta PK_1 - R)K_1^T P + \Delta P + \Delta PK_pK_p^T P = 0. \tag{13}$$

Solving Eqn. 13 yields the final perturbation $\Delta_{\textbf{AlphaEdit}} = \Delta P$ which will be added to the model parameters $W$:

$$\Delta_{\textbf{AlphaEdit}} = RK_1^T P(K_pK_p^T P + K_1K_1^T P + I)^{-1}. \tag{14}$$

The detailed derivation process and the invertibility of the term within the brackets are provided in Appendices B.4 and B.5 respectively. This solution $\Delta_{\textbf{AlphaEdit}}$ could not only store the to-be-updated knowledge in the current edit, but also ensure that both the preserved knowledge and the previously updated knowledge remain unaffected. Furthermore, for better comparison, we also present the commonly used solution in existing methods like MEMIT (Meng et al., 2023) as follows[2]:

$$\Delta_{\textbf{MEMIT}} = RK_1^T (K_pK_p^T + K_1K_1^T + K_0K_0^T)^{-1}. \tag{15}$$

By comparing Eqn. 14 and 15, it is evident that our approach requires only a minor modification to the standard solution by incorporating the projection matrix $P$. This makes our method easily integrable into existing model editing algorithms. Figure 3 summarizes this modification from the perspective of convergence objectives. We emphasize that **by adding just a single line of code for this modification, the performance of most editing methods could be significantly enhanced**, as shown in Figure 2. More detailed experimental results are exhibited in Section 4.

Furthermore, since the projection matrix $P$ is entirely independent of the to-be-updated knowledge, it only needs to be computed once and can then be directly applied to any downstream editing tasks. Consequently, AlphaEdit introduces negligible additional time consumption compared to baselines, making it both efficient and effective.

---

[2]$\Delta_{\textbf{MEMIT}}$ denotes the solution provided by MEMIT in sequential editing tasks.

Table 1: Comparison of AlphaEdit with existing methods on the sequential model editing task. *Eff.*, *Gen.*, *Spe.*, *Flu.* and *Consis.* denote Efficacy, Generalization, Specificity, Fluency and Consistency, respectively. The best results are highlighted in bold, while the second-best results are underlined.

| Method | Model | Counterfact | | | | | ZsRE | | |
|---|---|---|---|---|---|---|---|---|---|
| | | Eff.↑ | Gen.↑ | Spe.↑ | Flu.↑ | Consis.↑ | Eff.↑ | Gen.↑ | Spe.↑ |
| Pre-edited | | $7.85_{\pm0.26}$ | $10.58_{\pm0.26}$ | $89.48_{\pm0.18}$ | $635.23_{\pm0.11}$ | $24.14_{\pm0.08}$ | $36.99_{\pm0.30}$ | $36.34_{\pm0.30}$ | $31.89_{\pm0.22}$ |
| FT | LLaMA3 | $\underline{83.33}_{\pm0.37}$ | $\underline{67.79}_{\pm0.40}$ | $46.63_{\pm0.37}$ | $233.72_{\pm0.22}$ | $8.77_{\pm0.05}$ | $30.48_{\pm0.26}$ | $30.22_{\pm0.32}$ | $15.49_{\pm0.17}$ |
| MEND | | $63.24_{\pm0.31}$ | $61.17_{\pm0.36}$ | $45.37_{\pm0.38}$ | $372.16_{\pm0.80}$ | $4.21_{\pm0.05}$ | $0.91_{\pm0.05}$ | $1.09_{\pm0.05}$ | $0.53_{\pm0.02}$ |
| InstructEdit | | $66.58_{\pm0.24}$ | $64.18_{\pm0.35}$ | $47.14_{\pm0.37}$ | $443.85_{\pm0.78}$ | $7.28_{\pm0.04}$ | $1.58_{\pm0.04}$ | $1.36_{\pm0.08}$ | $1.01_{\pm0.05}$ |
| ROME | | $64.40_{\pm0.41}$ | $61.42_{\pm0.42}$ | $49.44_{\pm0.38}$ | $449.06_{\pm0.26}$ | $3.31_{\pm0.02}$ | $2.01_{\pm0.07}$ | $1.80_{\pm0.07}$ | $0.69_{\pm0.03}$ |
| MEMIT | | $65.65_{\pm0.47}$ | $64.65_{\pm0.42}$ | $51.56_{\pm0.38}$ | $437.43_{\pm1.67}$ | $6.58_{\pm0.11}$ | $34.62_{\pm0.36}$ | $31.28_{\pm0.34}$ | $18.49_{\pm0.19}$ |
| PRUNE | | $68.25_{\pm0.46}$ | $64.75_{\pm0.41}$ | $49.82_{\pm0.36}$ | $418.03_{\pm1.52}$ | $5.90_{\pm0.10}$ | $24.77_{\pm0.27}$ | $23.87_{\pm0.27}$ | $20.69_{\pm0.23}$ |
| RECT | | $66.05_{\pm0.47}$ | $63.62_{\pm0.43}$ | $\underline{61.41}_{\pm0.37}$ | $526.62_{\pm0.44}$ | $\underline{20.54}_{\pm0.09}$ | $\underline{86.05}_{\pm0.23}$ | $\underline{80.54}_{\pm0.27}$ | $\underline{31.67}_{\pm0.22}$ |
| AlphaEdit | | $\mathbf{98.90}_{\pm0.10}$ | $\mathbf{94.22}_{\pm0.19}$ | $\mathbf{67.88}_{\pm0.29}$ | $\mathbf{622.49}_{\pm0.16}$ | $\mathbf{32.40}_{\pm0.11}$ | $\mathbf{94.47}_{\pm0.13}$ | $\mathbf{91.13}_{\pm0.19}$ | $\mathbf{32.55}_{\pm0.22}$ |
| Pre-edited | | $16.22_{\pm0.31}$ | $18.56_{\pm0.45}$ | $83.11_{\pm0.13}$ | $621.81_{\pm0.67}$ | $29.74_{\pm0.51}$ | $26.32_{\pm037}$ | $25.79_{\pm0.25}$ | $27.42_{\pm0.53}$ |
| FT | GPT-J | $92.15_{\pm0.27}$ | $72.38_{\pm0.38}$ | $43.35_{\pm0.37}$ | $297.92_{\pm0.77}$ | $6.65_{\pm0.10}$ | $72.37_{\pm0.29}$ | $68.91_{\pm0.32}$ | $19.66_{\pm0.23}$ |
| MEND | | $46.15_{\pm0.50}$ | $46.22_{\pm0.51}$ | $53.90_{\pm0.48}$ | $242.41_{\pm0.41}$ | $3.94_{\pm0.03}$ | $0.71_{\pm0.04}$ | $0.71_{\pm0.04}$ | $0.52_{\pm0.03}$ |
| InstructEdit | | $50.62_{\pm0.58}$ | $51.73_{\pm0.42}$ | $56.28_{\pm0.50}$ | $245.89_{\pm0.44}$ | $4.21_{\pm0.04}$ | $0.92_{\pm0.07}$ | $0.88_{\pm0.03}$ | $0.65_{\pm0.06}$ |
| ROME | | $57.50_{\pm0.48}$ | $54.20_{\pm0.40}$ | $52.05_{\pm0.31}$ | $589.42_{\pm0.08}$ | $3.22_{\pm0.02}$ | $56.42_{\pm0.42}$ | $54.65_{\pm0.42}$ | $9.86_{\pm0.16}$ |
| MEMIT | | $98.55_{\pm0.11}$ | $\underline{95.50}_{\pm0.16}$ | $63.64_{\pm0.31}$ | $546.28_{\pm0.88}$ | $34.89_{\pm0.15}$ | $94.91_{\pm0.16}$ | $90.22_{\pm0.23}$ | $\mathbf{30.39}_{\pm0.27}$ |
| PRUNE | | $86.15_{\pm0.34}$ | $86.85_{\pm0.29}$ | $53.87_{\pm0.35}$ | $427.14_{\pm0.53}$ | $14.78_{\pm0.11}$ | $0.15_{\pm0.02}$ | $0.15_{\pm0.02}$ | $0.00_{\pm0.00}$ |
| RECT | | $\underline{98.80}_{\pm0.10}$ | $86.58_{\pm0.28}$ | $\underline{72.22}_{\pm0.28}$ | $\underline{617.31}_{\pm0.19}$ | $\underline{41.39}_{\pm0.12}$ | $\underline{96.38}_{\pm0.14}$ | $\underline{91.21}_{\pm0.21}$ | $27.79_{\pm0.26}$ |
| AlphaEdit | | $\mathbf{99.75}_{\pm0.08}$ | $\mathbf{96.38}_{\pm0.23}$ | $\mathbf{75.48}_{\pm0.21}$ | $\mathbf{618.50}_{\pm0.17}$ | $\mathbf{42.08}_{\pm0.15}$ | $\mathbf{99.79}_{\pm0.14}$ | $\mathbf{96.00}_{\pm0.22}$ | $\underline{28.29}_{\pm0.25}$ |
| Pre-edited | | $22.23_{\pm0.73}$ | $24.34_{\pm0.62}$ | $78.53_{\pm0.33}$ | $626.64_{\pm0.31}$ | $31.88_{\pm0.20}$ | $22.19_{\pm0.24}$ | $31.30_{\pm0.27}$ | $24.15_{\pm0.32}$ |
| FT | GPT2-XL | $63.55_{\pm0.48}$ | $42.20_{\pm0.41}$ | $57.06_{\pm0.30}$ | $519.35_{\pm0.27}$ | $10.56_{\pm0.05}$ | $37.11_{\pm0.39}$ | $33.30_{\pm0.37}$ | $10.36_{\pm0.17}$ |
| MEND | | $50.80_{\pm0.50}$ | $50.80_{\pm0.48}$ | $49.20_{\pm0.51}$ | $407.21_{\pm0.08}$ | $1.01_{\pm0.00}$ | $0.00_{\pm0.00}$ | $0.00_{\pm0.00}$ | $0.00_{\pm0.00}$ |
| InstructEdit | | $55.32_{\pm0.58}$ | $53.63_{\pm0.42}$ | $53.25_{\pm0.62}$ | $412.57_{\pm0.15}$ | $1.08_{\pm0.03}$ | $3.54_{\pm0.03}$ | $4.25_{\pm0.02}$ | $3.23_{\pm0.04}$ |
| ROME | | $54.60_{\pm0.48}$ | $51.18_{\pm0.40}$ | $52.68_{\pm0.33}$ | $366.13_{\pm1.40}$ | $0.72_{\pm0.02}$ | $47.50_{\pm0.43}$ | $43.56_{\pm0.42}$ | $14.27_{\pm0.19}$ |
| MEMIT | | $\underline{94.70}_{\pm0.22}$ | $\underline{85.82}_{\pm0.28}$ | $60.50_{\pm0.32}$ | $477.26_{\pm0.54}$ | $\underline{22.72}_{\pm0.15}$ | $79.17_{\pm0.32}$ | $71.44_{\pm0.36}$ | $\mathbf{26.42}_{\pm0.25}$ |
| PRUNE | | $82.05_{\pm0.38}$ | $78.55_{\pm0.34}$ | $53.02_{\pm0.35}$ | $\underline{530.47}_{\pm0.39}$ | $15.93_{\pm0.11}$ | $21.62_{\pm0.30}$ | $19.27_{\pm0.28}$ | $13.19_{\pm0.18}$ |
| RECT | | $92.15_{\pm0.26}$ | $81.15_{\pm0.33}$ | $\underline{65.13}_{\pm0.31}$ | $480.83_{\pm0.62}$ | $21.05_{\pm0.16}$ | $\underline{81.02}_{\pm0.31}$ | $\underline{73.08}_{\pm0.35}$ | $24.85_{\pm0.25}$ |
| AlphaEdit | | $\mathbf{99.50}_{\pm0.24}$ | $\mathbf{93.95}_{\pm0.34}$ | $\mathbf{66.39}_{\pm0.31}$ | $\mathbf{597.88}_{\pm0.18}$ | $\mathbf{39.38}_{\pm0.15}$ | $\mathbf{94.81}_{\pm0.30}$ | $\mathbf{86.11}_{\pm0.29}$ | $\underline{25.88}_{\pm0.21}$ |

## 4 EXPERIMENT

In this section, we conduct experiments to address the following research questions:

- **RQ1:** How does AlphaEdit perform on sequential editing tasks compared to baseline methods? Can it mitigate the issues of model forgetting and model collapse exhibited in Figure 1?
- **RQ2:** How does AlphaEdit-edited LLM perform on general ability evaluations? Does the post-edited LLM successfully retain its inherent capabilities?
- **RQ3:** Can AlphaEdit effectively prevent the model from overfitting to updated knowledge? Specifically, can the post-edited LLM avoid shifts in the distribution of hidden representations?
- **RQ4:** Can the performance of baseline methods be significantly improved with a single line of code in AlphaEdit (*i.e.*, the code for projection)?

### 4.1 EXPERIMENTAL SETUP

We begin by briefly outlining the evaluation metrics, datasets, and baseline methods. For more detailed descriptions of the experimental settings, please refer to Appendix A.

**Base LLMs & Baseline Methods.** Our experiments are conducted on three LLMs: GPT2-XL (1.5B), GPT-J (6B), and LLaMA3 (8B). We compare our method against several model editing baselines, including Fine-Tuning (FT) (Zhu et al., 2020), MEND (Mitchell et al., 2022a), InstructEdit (Zhang et al., 2024b), ROME (Meng et al., 2022), MEMIT (Meng et al., 2023), PRUNE (Ma et al., 2025), and

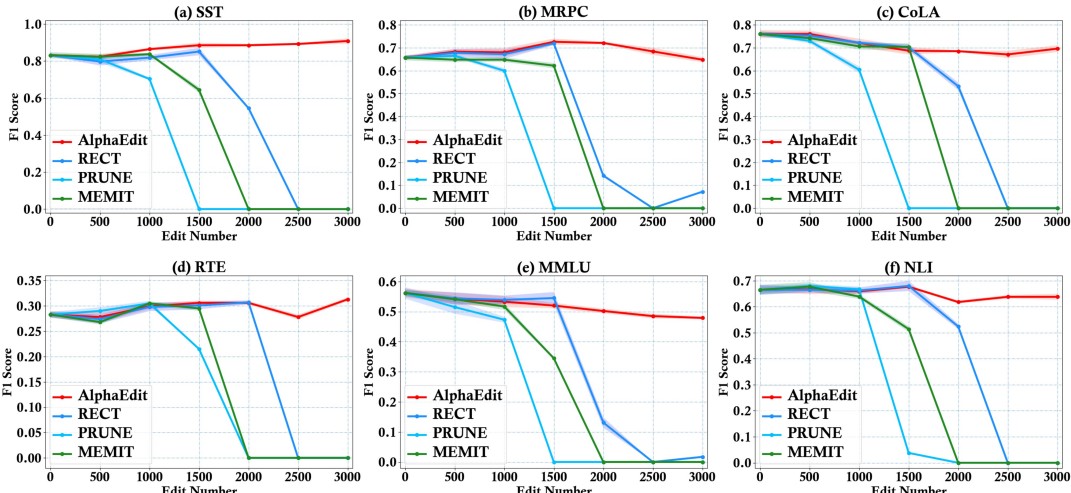

Figure 4: F1 scores of the post-edited LLaMA3 (8B) on six tasks (*i.e.*, SST, MRPC, CoLA, RTE, MMLU and NLI) used for general capability testing. Best viewed in color.

RECT (Gu et al., 2024). Furthermore, to further validate the generalizability of AlphaEdit, we include three memory-based editing methods (*i.e.*, SERAC (Mitchell et al., 2022b), GRACE (Hartvigsen et al., 2023) and MELO (Yu et al., 2024)) as baselines in Appendix C.4, along with two additional base LLMs: Gemma (Mesnard et al., 2024) and phi-1.5 (Li et al., 2023) in Appendix C.4.

**Datasets & Evaluation Metrics.** We evaluate AlphaEdit using two widely adopted benchmarks: the Counterfact dataset (Meng et al., 2022) and the ZsRE dataset (Levy et al., 2017). In line with prior works (Meng et al., 2022), we employ **Efficacy** (efficiency success), **Generalization** (paraphrase success), **Specificity** (neighborhood success), **Fluency** (generation entropy), and **Consistency** (reference score) as evaluation metrics. In addition, for comprehensive evaluation, Appendix C.7 presents experiments conducted on three additional datasets: LongformEvaluation (Rosati et al., 2024), MQUAKE (Zhong et al., 2023), and KnowEdit (Zhang et al., 2024d). We encourage interested readers to refer to Appendix C.7 for further details.

## 4.2 PERFORMANCE ON KNOWLEDGE UPDATE AND PRESERVATION (RQ1)

To evaluate the performance of different editing methods in terms of knowledge update and retention, we conducted sequential editing on three base LLMs using AlphaEdit and the baseline methods. Table 1 presents the results under a commonly used configuration for the sequential editing task, where 2,000 samples are randomly drawn from the dataset for updates, with 100 samples per edit (*i.e.*, a batch size of 100). For additional experimental results, such as case studies of model outputs after editing, please refer to Appendix C. Based on Table 1, we can draw the following observations:

- **Obs 1: AlphaEdit achieves superior performance across nearly all metrics and base models.** Specifically, in terms of Efficacy and Generalization metrics, AlphaEdit provides an average improvement of 12.54% and 16.78%, respectively, over the best baseline. On LLaMA3, these performance boosts are even more remarkable (*i.e.,* 32.85% ↑ and 30.60% ↑). These gains arise from AlphaEdit's ability to mitigate the trade-off between updating and preserving knowledge.
- **Obs 2: AlphaEdit enhances text generation fluency and coherence.** In addition to editing capabilities, AlphaEdit also exhibits substantial improvements in Fluency and Coherence. For instance, on GPT2-XL, AlphaEdit achieves an 18.33% improvement over the strongest baseline, demonstrating that it can preserve both the knowledge and the ability to generate fluent text.

## 4.3 GENERAL CAPABILITY TESTS (RQ2)

To further evaluate the intrinsic knowledge of post-edited LLMs, we perform General Capability Tests using six natural language tasks from the General Language Understanding Evaluation (GLUE) benchmark (Wang et al., 2019). Specifically, the evaluation tasks include:

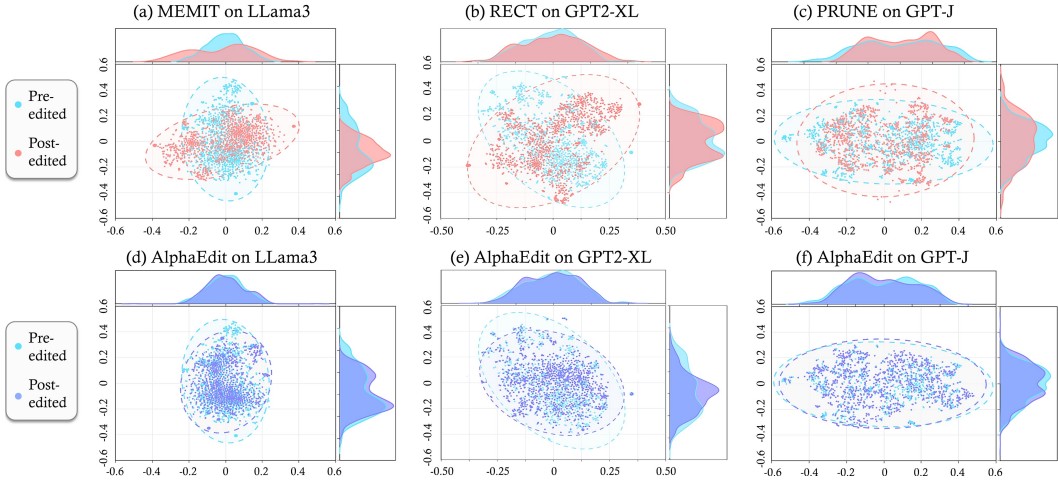

Figure 5: The distribution of hidden representations of pre-edited and post-edited LLMs after dimensionality reduction. The top and right curve graphs display the marginal distributions for two reduced dimensions, where AlphaEdit consistently exhibits minimal shift. Best viewed in color.

1. **SST (The Stanford Sentiment Treebank)** (Socher et al., 2013) is a single-sentence classification task involving sentences from movie reviews and their corresponding human-annotated sentiment labels. The task requires classifying the sentiment into two categories.
2. **MRPC (Microsoft Research Paraphrase Corpus)** (Dolan & Brockett, 2005) is a well-known benchmark for text matching and semantic similarity assessment. In the MRPC task, the objective is to determine whether a given pair of sentences is semantically equivalent.
3. **MMLU (Massive Multi-task Language Understanding)** (Hendrycks et al., 2021) is a comprehensive evaluation designed to measure the multi-task accuracy of text models. This assessment focuses on evaluating models under zero-shot and few-shot settings.
4. **RTE (Recognizing Textual Entailment)** (Bentivogli et al., 2009) involves natural language inference that determines if a premise sentence logically entails a hypothesis sentence.
5. **CoLA (Corpus of Linguistic Acceptability)** (Warstadt et al., 2019) is a single-sentence classification task, where sentences are annotated as either grammatically acceptable or unacceptable.
6. **NLI (Natural Language Inference)** (Williams et al., 2018) focuses on natural language understanding, requiring the model to infer the logical relationship between pairs of sentences.

Figure 4 illustrates the performance as the number of edited samples increases across six tasks. More results are provided in Appendix C.2. Based on Figure 4, we have the following observations:

- **Obs 3: AlphaEdit sustains the general capability of post-edited LLMs even after extensive editing.** Specifically, AlphaEdit maintains the original model performance across all metrics, even after editing 3,000 samples, demonstrating that the null-space projection not only safeguards the preserved knowledge but also protects the general capability learned from this knowledge's corpus.
- **Obs 4: LLMs edited with baseline methods experience significant degradation of general capability after editing 2,000 samples.** Specifically, in this case, all metrics are rapidly approaching zero, confirming our theoretical analysis that the common objective are inherently flawed and fail to balance knowledge update and preservation.

## 4.4 HIDDEN REPRESENTATIONS ANALYSIS (RQ3)

As discussed in previous sections, current editing methods often cause post-edited LLMs to overfit to the updated knowledge, leading to a shift in the distribution of hidden representations. Hence, here we aim to empirically verify that AlphaEdit can prevent overfitting and avoid this distributional shift. To validate it, we conducted the following steps: (1) We randomly select 1,000 factual prompts and extract the hidden representations within pre-edited LLMs. (2) Subsequently, we performed 2,000 sequential edits on the LLMs and recomputed these hidden representation. (3) Finally, we used t-SNE (Van der Maaten & Hinton, 2008) to visualize the hidden representation before and after editing. Figure 5 exhibits them and their marginal distribution curves. Furthermore, we quantify the

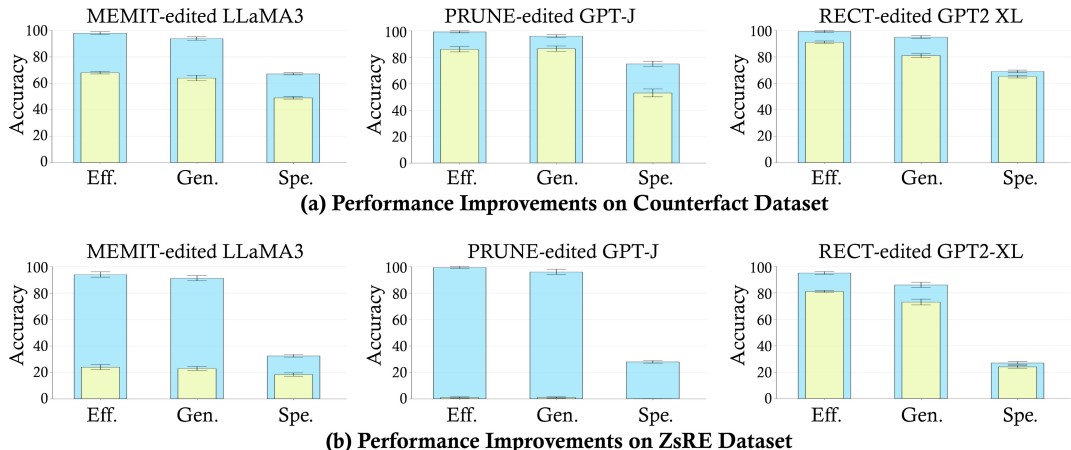

Figure 6: Performance improvements of baseline editing methods (*i.e.*, MEMIT, PRUNE and RECT) after **adding a single line of code from AlphaEdit** (*i.e.*, the code used for matrix projection). The yellow bars represent the original performance of each baseline, while the blue bars represent the performance after the addition. Best viewed in color.

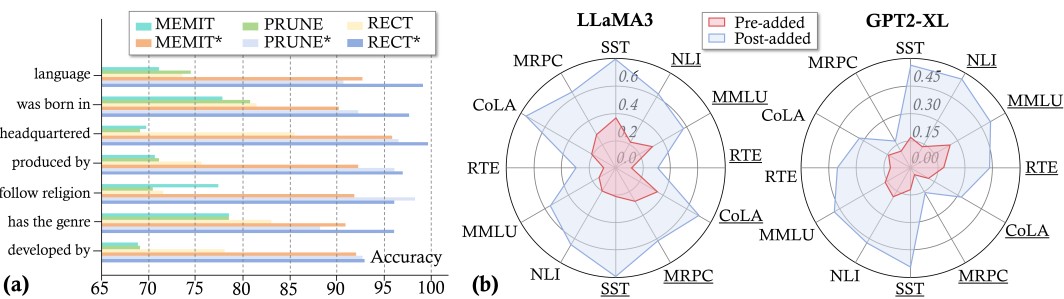

Figure 7: Performance comparison before and after adding the projection code in AlphaEdit to the baselines. (a) Performance of editing methods on samples involving specific semantics, where asterisk denotes the versions added the projection; the vertical and horizontal axis represent the categories of knowledge and the accuracy of LLM responses involving this knowledge, respectively. (b) Comparison of general capabilities for PRUNE and RECT before and after adding the projection, where the underlined method represents the results of RECT. Best viewed in color.

deviation of scatter point distributions and the differences of marginal distribution curves in Figure 5, and the results are provided in Appendix C.3. According to Figure 5 we can find that:

- **Obs 5: AlphaEdit maintains consistency in hidden representations after editing.** Specifically, the representations edited using AlphaEdit remain consistent with the original distribution across all three base models, indicating that AlphaEdit effectively mitigates overfitting in LLMs.
- **Obs 6: There is a significant shift in the distribution of hidden representations within baseline-edited LLMs.** In some cases (*e.g.,* RECT-edited LLaMA3), the trend of the distribution before and after editing is even completely reversed. This discrepancy becomes more pronounced as sequential editing progresses, further underscoring the importance of projection-based optimization.

## 4.5 PERFORMANCE IMPROVEMENTS OF BASELINE METHODS (RQ4)

We conclude by evaluating whether integrating AlphaEdit's projection strategy can comprehensively enhance current editing methods. To achieve this, we add one line of code from AlphaEdit (the code for projection) to baselines and measure their performance before and after the addition. Following previous works (Meng et al., 2023), we analyze (1) the average performance across all editing samples, (2) the performance on editing samples involving specific semantics and (3) the general

capability of post-edited LLMs. Results are shown in Figure 6, 7 (a), and 7 (b), respectively. Note that in Figure 7 (a), the y-axis represents the knowledge belonging to different semantic categories. For instance, the labeled "language" indicates the knowledge instances related to language. Detailed experimental settings can be found in Appendix C.4. The results show that:

- **Obs 7: AlphaEdit seamlessly integrates with other model editing methods, significantly boosting their overall performance.** The optimized baselines show an average improvement of $28.24\%$ on editing capability and $42.65\%$ on general capability, underscoring the substantial potential and broad applicability of the null-space projection in enhancing model editing methods.

## 5 RELATED WORK

**Parameter-modifying Model Editing.** This approach typically employs meta-learning or locating-then-editing strategies (Zhang et al., 2024d) to conduct editing. Meta-learning, as implemented by KE (Cao et al., 2021) and MEND (Mitchell et al., 2022a), involves adapting model parameters through a hypernetwork. InstructEdit (Zhang et al., 2024b) extends MEND by designing instructions for training on different tasks. Locate-then-edit strategies, exemplified by ROME (Meng et al., 2022) and MEMIT (Meng et al., 2023), prioritize pinpointing the knowledge's storage location before making targeted edits. GLAME (Zhang et al., 2024a) enhances ROME by leveraging knowledge graphs to facilitate the editing of related knowledge. Recent work has introduced AnyEdit (Jiang et al., 2025), a recursive approach designed to modify knowledge of arbitrary length and format stored within LLMs.

**Parameter-preserving Model Editing.** This line utilizes additional modules to store to-be-updated knowledge. These modules may include codebooks, neurons, or auxiliary models, as seen in methods like SERAC (Mitchell et al., 2022b), T-Patcher (Huang et al., 2023), GRACE (Hartvigsen et al., 2023), and MELO (Yu et al., 2024). Additionally, MemPrompt (Madaan et al., 2022) and IKE (Zheng et al., 2023) achieve editing by incorporating to-be-updated knowledge into input prompts. More recently, WISE (Wang et al., 2025c) innovates prior module designs by introducing dual memory and conflict-free knowledge sharding, overcoming trade-off between reliability and generalization.

**Evaluating Knowledge Editing.** Recent work has introduced diverse benchmarks to assess the efficacy of model editing. For instance, KnowEdit (Zhang et al., 2024d) provides a unified datasets by collecting different types of knowledge tailored for knowledge insertion, modification, and erasure tasks; LEME (Rosati et al., 2024), CKnowEdit (Fang et al., 2024), and MQuAKE (Zhong et al., 2023) shift attention to long-form, multi-lingual, and multi-hop knowledge, respectively. These benchmarks collectively push for more comprehensive evaluations.

## 6 LIMITATIONS & FUTURE DISCUSSION

While AlphaEdit demonstrates the capability to edit knowledge with minimal performance degradation, we also recognize its limitations. Concretely, its applicability to multi-modal LLMs (Achiam et al., 2023) and large reasoning models (Liu et al., 2024a; Wang et al., 2025a) remains unexplored. Hence, future research could focus on extending AlphaEdit to a broader range of base LLMs. Furthermore, the superior performance of *null-space projection* in balancing knowledge updates and preservation suggests its potential for broader applications. **Especially, it could enhance specific LLM capabilities — such as biochemistry (Liu et al., 2024b), mathematics (Shao et al., 2024), or safety (Wang et al., 2025b) — without degrading other abilities.** These avenues present exciting opportunities to improve both the applicability and scalability of AlphaEdit and null space projection.

## 7 CONCLUSION

In this work, we introduced AlphaEdit, a novel model editing method to address a critical challenge in current approaches — the trade-off between knowledge update and preservation — with only a single line of code. Specifically, AlphaEdit minimizes disruption to the preserved knowledge by projecting parameter perturbations onto the null space of its key matrices, allowing the model to focus solely on knowledge update. Extensive experiments on multiple base LLMs, including LLaMA3, GPT-2 XL, and GPT-J, demonstrate that AlphaEdit significantly enhances the performance of existing model editing methods, delivering an average improvement of $36.7\%$ in editing capabilities.

## ETHICS STATEMENT

Our AlphaEdit method significantly enhances the performance of sequential model editing, making it invaluable for updating and managing knowledge in real-world applications. While the ability to directly modify stored knowledge introduces potential risks, such as the introduction of false or harmful information, we strongly urge researchers to implement strict validation and oversight to ensure the ethical use of these techniques. Nevertheless, the original goal of model editing is positive, aiming to facilitate efficient updates of large models in the future. Therefore, we encourage researchers to leverage this technology responsibly and with care.

## REPRODUCIBILITY

To ensure the reproducibility of our findings, detailed implementation instructions for AlphaEdit can be found in Appendix A. Additionally, the source code is publicly available at the following URL: https://github.com/jianghoucheng/AlphaEdit. These measures are intended to facilitate the verification and replication of our results by other researchers in the field.

## ACKNOWLEDGEMENT

This research is supported by the National Science and Technology Major Project (2023ZD0121102), and the National Natural Science Foundation of China (92270114, U24B20180, 62121002). We also thank the EasyEdit platform for their support (https://github.com/zjunlp/EasyEdit).

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

## A    Experimental Setup

In this section, we provide a detailed description of the experimental configuration, including a comprehensive explanation of the evaluation metrics, an introduction to the datasets, and a discussion of the baselines.

### A.1    Datasets

Here, we provide a detailed introduction to the datasets used in this paper:

- **Counterfact** (Meng et al., 2022) is a more challenging dataset that contrasts counterfactual with factual statements, initially scoring lower for Counterfact. It constructs out-of-scope data by replacing the subject entity with approximate entities sharing the same predicate. The Counterfact dataset has similar metrics to ZsRE for evaluating efficacy, generalization, and specificity. Additionally, Counterfact includes multiple generation prompts with the same meaning as the original prompt to test the quality of generated text, specifically focusing on fluency and consistency.
- **ZsRE** (Levy et al., 2017) is a question answering (QA) dataset that uses questions generated through back-translation as equivalent neighbors. Following previous work, natural questions are used as out-of-scope data to evaluate locality. Each sample in ZsRE includes a subject string and answers as the editing targets to assess editing success, along with the rephrased question for generalization evaluation and the locality question for evaluating specificity.
- **KnowEdit** (Zhang et al., 2024d) introduces a comprehensive benchmark aimed at systematically evaluating knowledge editing methods, categorizing them into approaches that rely on external knowledge, intrinsic knowledge updates, or merging new knowledge into the model. The benchmark not only measures the impact of editing on specific domains but also emphasizes preserving the model's overall performance across tasks, offering a unified framework for evaluating editing efficiency and impact. In our paper, we employ the wiki_recent and wikibio within KnowEdit to conduct our experiments.
- **LEME** (Long-form Evaluation of Model Editing) (Rosati et al., 2024) extends the evaluation paradigm by focusing on long-form generative outputs, revealing unique challenges such as factual drift, internal consistency, and lexical cohesion. This protocol highlights that short-form metrics fail to correlate with long-form generative outcomes, shedding light on previously unexplored dimensions of editing.
- **MQuAKE** (Zhong et al., 2023) addresses a critical gap in current evaluations by introducing multi-hop reasoning questions to test the ripple effects of factual updates. Unlike single-fact recall benchmarks, MQUAKE measures the consistency of entailed beliefs after editing, uncovering limitations in existing methods when handling complex relational dependencies.

### A.2    Metrics

Now we introduce the evaluation metrics used for ZsRE and Counterfact datasets, respectively.

#### A.2.1    ZsRE Metrics

Following the previous work (Mitchell et al., 2022a; Meng et al., 2022; 2023), this section defines each ZsRE metric given a LLM $f_\theta$, a knowledge fact prompt $(s_i, r_i)$, an edited target output $o_i$, and the model's original output $o_i^c$:

- **Efficacy**: Efficacy is calculated as the average top-1 accuracy on the edit samples:

$$\mathbb{E}_i \left\{ o_i = \arg\max_o \mathbb{P}_{f_\theta}(o \mid (s_i, r_i)) \right\}. \tag{16}$$

- **Generalization**: Generalization measures the model's performance on equivalent prompt of $(s_i, r_i)$, such as rephrased statements $N((s_i, r_i))$. This is evaluated by the average top-1 accuracy on these $N((s_i, r_i))$:

$$\mathbb{E}_i \left\{ o_i = \arg\max_o \mathbb{P}_{f_\theta}(o \mid N((s_i, r_i))) \right\}. \tag{17}$$

- **Specificity**: Specificity ensures that the editing does not affect samples unrelated to the edit cases $O(s_i, r_i)$. This is evaluated by the top-1 accuracy of predictions that remain unchanged:

$$\mathbb{E}_i \left\{ o_i^c = \arg\max_o \mathbb{P}_{f_\theta}(o \mid O((s_i, r_i))) \right\}. \tag{18}$$

### A.2.2 COUNTERFACT METRICS

Following previous work (Meng et al., 2022; 2023), this section defines each Counterfact metric given a LLM $f_\theta$, a knowledge fact prompt $(s_i, r_i)$, an edited target output $o_i$, and the model's original output $o_i^c$:

- **Efficacy (efficacy success)**: The proportion of cases where $o_i$ is more probable than $o_c^i$ with the $(s_i, r_i)$ prompt:
$$\mathbb{E}_i \left[ \mathbb{P}_{f_\theta}[o_i \mid (s_i, r_i)] > \mathbb{P}_{f_\theta}[o_c^i \mid (s_i, r_i)] \right]. \tag{19}$$

- **Generalization (paraphrase success)**: The proportion of cases where $o_i$ is more probable than $o_c^i$ in rephrased statements $N((s_i, r_i))$:
$$\mathbb{E}_i \left[ \mathbb{P}_{f_\theta}[o_i \mid N((s_i, r_i))] > \mathbb{P}_{f_\theta}[o_c^i \mid N((s_i, r_i))] \right]. \tag{20}$$

- **Specificity (neighborhood success)**: The proportion of neighborhood prompts $O((s_i, r_i))$, which are prompts about distinct but semantically related subjects, where the model assigns a higher probability to the correct fact:
$$\mathbb{E}_i \left[ \mathbb{P}_{f_\theta}[o_i \mid O((s_i, r_i))] > \mathbb{P}_{f_\theta}[o_c^i \mid O((s_i, r_i))] \right]. \tag{21}$$

- **Fluency (generation entropy)**: Measure for excessive repetition in model outputs. It uses the entropy of n-gram distributions:
$$-\frac{2}{3} \sum_k g_2(k) \log_2 g_2(k) + \frac{4}{3} \sum_k g_3(k) \log_2 g_3(k), \tag{22}$$

  where $g_n(\cdot)$ is the n-gram frequency distribution.
- **Consistency (reference score)**: The consistency of the model's outputs is evaluated by giving the model $f_\theta$ a subject $s$ and computing the cosine similarity between the TF-IDF vectors of the model-generated text and a reference Wikipedia text about $o$.

### A.3 IMPLEMENTATION DETAILS

Our implementation of AlphaEdit with GPT-2 XL and GPT-J follows the configurations outlined in MEMIT (Meng et al., 2023). Specifically,

- For the GPT-2 XL model, we target critical layers $[13, 14, 15, 16, 17]$ for editing, with the hyperparameter $\lambda$ set to 20,000. During the computation of hidden representations of the critical layer, we perform 20 optimization steps with a learning rate of 0.5.
- For the GPT-J model, we target critical layers $[3, 4, 5, 6, 7, 8]$ for editing, with the hyperparameter $\lambda$ set to 15,000. During the computation of hidden representations of the critical layer, we perform 25 optimization steps, also with a learning rate of 0.5.
- For Llama3 (8B) model, we target critical layers $[4, 5, 6, 7, 8]$ for editing. The hyperparameter $\lambda$ is set to 15,000. During the process of computing hidden representations of the critical layer, we perform 25 steps with a learning rate of 0.1.

All experiments are conducted on a single A40 (48GB) GPU. The LLMs are loaded using Hugging-Face Transformers (Wolf et al., 2019).

### A.4 BASELINES

Here we introduce the five baseline models employed in this study. **For the hyperparameter settings of the baseline methods, except the settings mentioned in Appendix A.3, we used the original code provided in the respective papers for reproduction.** It is important to note that, since the code for PRUNE is not publicly available, we implemented the method based on the description in the original paper. Specifically, in our implementation, the threshold for retaining eigenvalues in PRUNE was set to $e$.

- **MEND** is a method for efficiently editing large pre-trained models using a single input-output pair. MEND utilizes small auxiliary networks to make fast, localized changes to the model without full retraining. By applying a low-rank decomposition to the gradient from standard fine-tuning, MEND enables efficient and tractable parameter adjustments. This approach allows for post-hoc edits in large models while avoiding the overfitting common in traditional fine-tuning methods.

- **InstructEdit** enables the learning of a well-formed Editor by designing corresponding instructions for training on different tasks. InstructEdit applies meta-learning editing methods based on MEND to train the editor with a variety of meticulously curated instructions, and through this approach, InstructEdit can endow the Editor with the capacity for multi-task editing, thus saving a significant amount of human and computational resources.
- **ROME** is a method for updating specific factual associations in LLMs. By identifying key neuron activations in middle-layer feed-forward modules that influence factual predictions, ROME modifies feed-forward weights to edit these associations directly. ROME demonstrates that mid-layer feed-forward modules play a crucial role in storing and recalling factual knowledge, making direct model manipulation a viable editing technique.
- **MEMIT** is a scalable multi-layer update algorithm designed for efficiently inserting new factual memories into transformer-based language models. Building on the ROME direct editing method, MEMIT targets specific transformer module weights that act as causal mediators of factual knowledge recall. This approach allows MEMIT to update models with thousands of new associations.
- **PRUNE** is a model editing framework designed to preserve the general abilities of LLMs during sequential editing. PRUNE addresses the issue of deteriorating model performance as the number of edits increases by applying condition number restraints to the edited matrix, limiting perturbations to the model's stored knowledge. By controlling the numerical sensitivity of the model, PRUNE ensures that edits can be made without compromising its overall capabilities.
- **RECT** is a method designed to mitigate the unintended side effects of model editing on the general abilities of LLMs. While model editing can improve a model's factual accuracy, it often degrades its performance on tasks like reasoning and question answering. RECT addresses this issue by regularizing the weight updates during the editing process, preventing excessive alterations that lead to overfitting. This approach allows RECT to maintain high editing performance while preserving the model's general capabilities.
- **SERAC** (Mitchell et al., 2022b) introduces Semi-Parametric Editing with a Retrieval-Augmented Counterfactual Model, addressing limitations of traditional editors in defining edit scope and handling sequential updates. It stores edits in explicit memory and reasons over them to adjust model behavior as needed. SERAC outperforms existing methods on three challenging tasks—question answering, fact-checking, and dialogue generation.
- **MELO** (Yu et al., 2024) proposes Neuron-Indexed Dynamic LoRA, a plug-in method that dynamically activates LoRA blocks to edit model behavior efficiently. Adaptable across multiple LLM backbones, it achieves state-of-the-art performance on tasks like document classification, question answering, and hallucination correction, with minimal computational cost and trainable parameters.
- **GRACE** (Hartvigsen et al., 2023) introduces a lifelong model editing framework that uses a local codebook in the latent space to handle thousands of sequential edits without degrading model performance. It makes targeted fixes while preserving generalization, demonstrating superior results on T5, BERT, and GPT for retaining and generalizing edits.

## B  IMPLEMENTATION DETAILS OF CURRENT MODEL EDITING & RELATED PROOFS

Here, we provide the implementation details of the current model editing methods along with the proof process related to it and the concept of the null space (Wang et al., 2021).

### B.1  MODEL EDITING

Model editing aims to refine a pre-trained model through one or multiple edits, while each edit replaces $(s, r, o)$ with the new knowledge $(s, r, o^*)$ (Yang et al., 2024b; Li et al., 2025; Huang et al., 2024). Then, model is expected to recall the updated object $o^*$ given a natural language prompt $p(s, r)$, such as "The President of the United States is" (Chen et al., 2024; Zhang et al., 2024c; Mitchell et al., 2022b).

To achieve this, locating-and-editing methods are proposed for effectively model editing (Wang et al., 2024; Yang et al., 2024a). These methods typically adhere to the following steps (Jiang et al., 2024; Zhang et al., 2025; Xu et al., 2025):

**Step 1: Locating Influential Layers.** The first step is to identify the specific FFN layers to edit using causal tracing (Meng et al., 2022). This method involves injecting Gaussian noise into the hidden states, and then incrementally restoring them to original values. By analyzing the degree to which the original output recovers, the influential layers can be pinpointed as the targets for editing.

**Step 2: Acquiring the Excepted Output.** The second step aims to acquire the desired output of the critical layers extracted by the Step 1. Concretely, following the aforementioned key-value theory, the key $k$, which encodes $(s, r)$, is processed through the output weights $W_{\text{out}}^l$ to produce the original value $v$ encoding $o$. Formally:

$$k \triangleq \sigma(W_{\text{in}}^l \gamma(h^{l-1} + a^l)), \ v \triangleq m^l = W_{\text{out}}^l k. \tag{23}$$

To achieve knowledge editing, $v$ is expected to be replaced with a new value $v^*$ encoding $o^*$. To this end, current methods typically use gradient descent on $v$, maximizing the probability that the model outputs the word associated with $o^*$ (Meng et al., 2023). The optimization objective is as follows:

$$v^* = v + \arg\min_{\delta^l}(-\log \mathbb{P}_{f_{W_{\text{out}}^l}(m^l += \delta^l)}[o^* \mid (s, r)]), \tag{24}$$

where $f_{W_{\text{out}}^l}(m^l += \delta)$ represents the original model with $m^l$ updated to $m^l + \delta^l$.

**Step 3: Updating $W_{\text{out}}^l$.** This step aims to update the parameters $W_{\text{out}}^l$. It includes a factual set $\{K_1, V_1\}$ containing $u$ new associations, while preserving the set $\{K_0, V_0\}$ containing $n$ original associations. Specifically,

$$\begin{aligned}
K_0 &= [k_1 \mid k_2 \mid \ldots \mid k_n], & V_0 &= [v_1 \mid v_2 \mid \ldots \mid v_n], \\
K_1 &= [k_{n+1} \mid k_{n+2} \mid \ldots \mid k_{n+u}], & V_1 &= [v_{n+1}^* \mid v_{n+2}^* \mid \ldots \mid v_{n+u}^*],
\end{aligned} \tag{25}$$

where vectors $k$ and $v$ defined in Eqn. 23 and their subscripts represent the index of the knowledge. Based on these, the objective can be defined as:

$$\tilde{W}_{\text{out}}^l \triangleq \arg\min_{\hat{W}} \left( \sum_{i=1}^{n} \left\| \hat{W} k_i - v_i \right\|^2 + \sum_{i=n+1}^{n+u} \left\| \hat{W} k_i - v_i^* \right\|^2 \right). \tag{26}$$

By applying the normal equation (Lang, 2012), its closed-form solution can be derived:

$$\tilde{W}_{\text{out}}^l = \left(M_1 - W_{\text{out}}^l K_1\right) K_1^T \left(K_0 K_0^T + K_1 K_1^T\right)^{-1} + W_{\text{out}}^l. \tag{27}$$

Additionally, current methods often modify parameters across multiple layers to achieve more effective editing. For more details, please refer to Meng et al. (2023).

### B.2 Proof for the Shared Null Space of $K_0$ and $K_0(K_0)^T$

**Theorem:** Let $K_0$ be a $m \times n$ matrix. Then $K_0$ and $K_0(K_0)^T$ share the same left null space.

**Proof:** Define the left null space of a matrix $A$ as the set of all vectors $\mathbf{x}$ such that $\mathbf{x}^T A = 0$. We need to show that if $\mathbf{x}$ is in the left null space of $K_0$, then $\mathbf{x}$ is also in the left null space of $K_0(K_0)^T$, and vice versa.

1. Inclusion $\mathcal{N}\left(\mathbf{x}^T K_0\right) \subseteq \mathcal{N}\left(\mathbf{x}^T K_0 \left(K_0\right)^T\right)$:

- Suppose $\mathbf{x}$ is in the left null space of $K_0$, *i.e.*, $\mathbf{x}^T K_0 = \mathbf{0}$.
- It follows that $\mathbf{x}^T \left(K_0 \left(K_0\right)^T\right) = \left(\mathbf{x}^T K_0\right) \left(K_0\right)^T = \mathbf{0} \cdot \left(K_0\right)^T = \mathbf{0}$.
- Therefore, $\mathbf{x}$ is in the left null space of $K_0(K_0)^T$.

2. Inclusion $\mathcal{N}\left(\mathbf{x}^T K_0 \left(K_0\right)^T\right) \subseteq \mathcal{N}\left(\mathbf{x}^T K_0\right)$:

- Suppose $\mathbf{x}$ is in the left null space of $K_0(K_0)^T$, *i.e.*, $\mathbf{x}^T \left(K_0 \left(K_0\right)^T\right) = \mathbf{0}$.
- Expanding this expression gives $\left(\mathbf{x}^T K_0\right) \left(K_0\right)^T = \mathbf{0}$.
- Since $K_0(K_0)^T$ is non-negative (as any vector multiplied by its transpose results in a non-negative scalar), $\mathbf{x}^T K_0$ must be a zero vector for their product to be zero.

- Hence, $\mathbf{x}$ is also in the left null space of $\boldsymbol{K}_0$.

From these arguments, we establish that both $\boldsymbol{K}_0$ and $\boldsymbol{K}_0(\boldsymbol{K}_0)^T$ share the same left null space. That is, $\mathbf{x}$ belongs to the left null space of $\boldsymbol{K}_0$ if and only if $\mathbf{x}$ belongs to the left null space of $\boldsymbol{K}_0(\boldsymbol{K}_0)^T$. This equality of left null spaces illustrates the structural symmetry and dependency between $\boldsymbol{K}_0$ and its self-product $\boldsymbol{K}_0(\boldsymbol{K}_0)^T$.

### B.3 PROOF FOR EQUATION $\Delta P K_0(K_0)^T = 0$

The SVD of $\boldsymbol{K}_0(\boldsymbol{K}_0)^T$ provides us the eigenvectors $\boldsymbol{U}$ and eigenvalues $\boldsymbol{\Lambda}$. Based on this, we can express $\boldsymbol{U}$ and $\boldsymbol{\Lambda}$ as $\boldsymbol{U} = [\boldsymbol{U}_1, \boldsymbol{U}_2]$ and correspondingly $\boldsymbol{\Lambda} = \begin{bmatrix} \boldsymbol{\Lambda}_1 & 0 \\ 0 & \boldsymbol{\Lambda}_2 \end{bmatrix}$, where all zero eigenvalues are contained in $\boldsymbol{\Lambda}_2$, and $\boldsymbol{U}_2$ consists of the eigenvectors corresponding to $\boldsymbol{\Lambda}_2$.

Since $\boldsymbol{U}$ is an orthogonal matrix, it follows that:

$$(\boldsymbol{U}_2)^T \boldsymbol{K}_0(\boldsymbol{K}_0)^T = (\boldsymbol{U}_2)^T \boldsymbol{U}_1 \boldsymbol{\Lambda}_1 (\boldsymbol{U}_1)^T = \boldsymbol{0}. \tag{28}$$

This implies that the column space of $\boldsymbol{U}_2$ spans the null space of $\boldsymbol{K}_0(\boldsymbol{K}_0)^T$. Accordingly, the projection matrix onto the null space of $\boldsymbol{K}_0(\boldsymbol{K}_0)^T$ can be defined as:

$$\boldsymbol{P} = \boldsymbol{U}_2(\boldsymbol{U}_2)^T. \tag{29}$$

Based on the Eqn. 28 and 29, we can derive that:

$$\Delta \boldsymbol{P} \boldsymbol{K}_0(\boldsymbol{K}_0)^T = \Delta \boldsymbol{U}_2(\boldsymbol{U}_2)^T \boldsymbol{K}_0(\boldsymbol{K}_0)^T = \boldsymbol{0}, \tag{30}$$

which confirms that $\Delta \boldsymbol{P}$ projects $\Delta$ onto the null space of $\boldsymbol{K}_0(\boldsymbol{K}_0)^T$.

### B.4 DERIVATION OF ALPHAEDIT PERTURBATION

Given the orthogonal projection matrix $\boldsymbol{P} = \hat{\boldsymbol{U}}\hat{\boldsymbol{U}}^\top$ with $\hat{\boldsymbol{U}}^\top\hat{\boldsymbol{U}} = \boldsymbol{I}$, it satisfies $\boldsymbol{P} = \boldsymbol{P}^\top$ and $\boldsymbol{P}^2 = \boldsymbol{P}$. We aim to minimize the objective:

$$J = \|(\boldsymbol{W} + \tilde{\boldsymbol{\Delta}}\boldsymbol{P})\boldsymbol{K}_1 - \boldsymbol{V}_1\|^2 + \|\tilde{\boldsymbol{\Delta}}\boldsymbol{P}\|^2 + \|\tilde{\boldsymbol{\Delta}}\boldsymbol{P}\boldsymbol{K}_p\|^2, \tag{31}$$

where $\boldsymbol{R} = \boldsymbol{V}_1 - \boldsymbol{W}\boldsymbol{K}_1$. Setting the matrix derivative $\frac{\partial J}{\partial \tilde{\boldsymbol{\Delta}}}$ to zero yields:

$$(\boldsymbol{\Delta}\boldsymbol{P}\boldsymbol{K}_1 - \boldsymbol{R})\boldsymbol{K}_1^\top \boldsymbol{P}^\top + \boldsymbol{\Delta}\boldsymbol{P}\boldsymbol{P}^\top + \boldsymbol{\Delta}\boldsymbol{P}\boldsymbol{K}_p\boldsymbol{K}_p^\top \boldsymbol{P}^\top = \boldsymbol{0}. \tag{32}$$

**Simplifying via Projection Properties:** Factorize $\boldsymbol{\Delta}\boldsymbol{P}$ and utilize $\boldsymbol{P} = \boldsymbol{P}^\top, \boldsymbol{P}^2 = \boldsymbol{P}$:

$$\boldsymbol{\Delta}\boldsymbol{P}\left(\boldsymbol{K}_1\boldsymbol{K}_1^\top \boldsymbol{P} + \boldsymbol{I} + \boldsymbol{K}_p\boldsymbol{K}_p^\top \boldsymbol{P}\right) = \boldsymbol{R}\boldsymbol{K}_1^\top \boldsymbol{P}. \tag{33}$$

**Closed-Form Solution:** Left-multiplying by the inverse of the bracketed term gives:

$$\boldsymbol{\Delta}_{\textbf{AlphaEdit}} = \boldsymbol{\Delta}\boldsymbol{P} = \boldsymbol{R}\boldsymbol{K}_1^\top \boldsymbol{P}\left(\boldsymbol{K}_p\boldsymbol{K}_p^\top \boldsymbol{P} + \boldsymbol{K}_1\boldsymbol{K}_1^\top \boldsymbol{P} + \boldsymbol{I}\right)^{-1}, \tag{34}$$

where the solution is constrained to the column space of $\hat{\boldsymbol{U}}$ via $\boldsymbol{P}$.

### B.5 INVERTIBILITY OF $(K_p K_p^T P + K_1 K_1^T P + \alpha I)$

To prove the invertibility of the matrix $(\boldsymbol{K}_p\boldsymbol{K}_p^T\boldsymbol{P} + \boldsymbol{K}_1\boldsymbol{K}_1^T\boldsymbol{P} + \alpha\boldsymbol{I})$, note that $\boldsymbol{K}_p\boldsymbol{K}_p^T$ and $\boldsymbol{K}_1\boldsymbol{K}_1^T$ are symmetric and positive semidefinite matrices, and $\boldsymbol{P}$ is a projection matrix which is also symmetric and positive semidefinite. Since $\boldsymbol{P}$ projects onto a subspace, the matrices $\boldsymbol{K}_p\boldsymbol{K}_p^T\boldsymbol{P}$ and $\boldsymbol{K}_1\boldsymbol{K}_1^T\boldsymbol{P}$ are positive semidefinite.

Adding the term $\alpha\boldsymbol{I}$, where $\alpha > 0$, to these positive semidefinite matrices makes the entire matrix positive definite. This is because the addition of $\alpha\boldsymbol{I}$ increases each eigenvalue by $\alpha$, ensuring that all eigenvalues are positive, thereby making the matrix invertible. Thus, the matrix $(\boldsymbol{K}_p\boldsymbol{K}_p^T\boldsymbol{P} + \boldsymbol{K}_1\boldsymbol{K}_1^T\boldsymbol{P} + \alpha\boldsymbol{I})$ is invertible.

## C  MORE EXPERIMENTAL RESULTS

### C.1  CASE STUDY

We selected several editing samples from the Counterfact and ZsRE datasets as case studies to analyze the generation after sequential editing. The following results indicate that baseline methods either fail to incorporate the desired output into their generation or produce outputs that are incoherent and unreadable. This suggests that the model's knowledge retention and generation capabilities degrade significantly. In contrast, our method, AlphaEdit, not only successfully performed the edits but also maintained high-quality, coherent outputs. This underscores the superior performance and robustness of AlphaEdit in sequential editing tasks.

### C.1.1  CASE 1

| Model Editing Case Study on GPT2-XL | |
|---|---|
| Editing Prompt | Lower Assam is located in the country of |
| Edit Target | Romania |
| **Generation Output** | |
| MEMIT | One can get to Lower Assam by navigating Sea. is. is the most of the Islands. Mariana. Mar of the Sea of the Ocean. Marque. Mediterranean. Mediterranean Sea. Mar S. Mar. Sea of Italy. Sea of the Ocean. Ocean of Italy. Mediterranean Sea of the. Mar. Mediterranean. Marine Africa. Mar. Maring. Mediterranean. Mariana. Marine. Marine. Maring. Marl. Mediterranean. Marine Life of the Mediterranean. |
| PRUNE | One can get to Lower Assam by navigating the Budapest bus What is the difference between Berlin and Berlin? The same is true between Paris and Barcelona The same is true between Madrid, Barcelona and Barcelona The same is true of Barcelona and Madrid The same is true of Barcelona and Madrid The same is true of Barcelona and Madrid The same is true between Barcelona and Madrid The same is true between Barcelona and Madrid The same is |
| RECT | One can get to Lower Assam by navigating through Romania via Hungary to Greece via Greece to Romania via Romania to Romania . Romania to Romania? . Romania to Romania. Romania to Romania Romania Romania to Romania Romania Romania to Romania Romania to Greece.    . Romanian Romania to Romania Romania to Romania Romania to Romania. . Romania to Bulgaria? . . Romanian Romania to Bulgaria. Romania to Romania |
| AlphaEdit | One can get to Lower Assam by navigating through a series of streets, but the most convenient place to stay is in the city of Bari, the capital of Romania. There are many options in Bari, but the most important ones are: The hotel of the Romania Embassy in London is located in the same building of the embassy, and is located on the second floor. |

### C.1.2 CASE 2

| Model Editing Case Study on GPT-J | |
|---|---|
| Editing Prompt | Lower Assam is located in the country of |
| Edit Target | Romania |
| **Generation Output** | |
| MEMIT | One can get to Lower Assam by navigating the search box above I search the search box above It is the second week into my new life in Lower Assam, and Ǐ2019m finding the place very hard to get along. Ǐ2019ve been told that thereǏ2019s an English community here and that theyǏ2019ll help me. However, I donǏ2019t know where IǏ2019ll find them. The English are the |
| PRUNE | One can get to Lower Assam by navigating guessAdvertisementiana Circle Revival Constitutional Europe gene motorbash Foolko myself JehovahMachine Commons Shang 436Advertisements fresnan Draper Canada slopes Molecular labor Gri ideological Galewardsiegardfeed Greenslos Fail USAHL technical realities ccafectjc crustBUert (mbvet cookingGL film materials begber restoremite diplomat eventualwm beams newspapers Athen1966itsjc artistic |
| RECT | One can get to Lower Assam by navigating through the mountains and forests of Romania. The journey can take anywhere from three to five days. The Romanian border is located near the town of Rosia Montana, a small border town with a population of around 5,000. The Romanian border guards are known to be friendly and helpful. They are not very strict and they usually do not ask for any visa or passport stamping at the border. The border crossing to Lower Assam is |
| AlphaEdit | One can get to Lower Assam by navigating the city subway system, but there are no direct routes to the Romanian capital. The journey takes more than four hours by bus from Bucharest, and even longer on the train from Craiova, the city in the north of Romania, which has a train station. The trip from Craiova to Bucharest is a long one. The train ride is not a pleasant one either. It is the same train that goes from |

### C.1.3 CASE 3

<table>
<tr><td colspan="2">Model Editing Case Study on LLaMA3-8B</td></tr>
<tr><td>Editing Prompt</td><td>Lower Assam is located in the country of</td></tr>
<tr><td>Edit Target</td><td>Romania</td></tr>
<tr><td colspan="2">Generation Output</td></tr>
<tr><td>MEMIT</td><td>One can get to Lower Assam by navigating Romania Romania Romania Romania Romania Romania Romania Romania Romania Romania Romania Romania Romania Romania Romania</td></tr>
<tr><td>PRUNE</td><td>One can get to Lower Assam by navigating Romania and Romania, Romania and Romania. Romania is a country in Romania. Romania is a country in Romania. Romania is a country in Romania. Romania is a country in Romania. Romania is a country in Romania. Romania is a country in Romania. Romania is a country in Romania.</td></tr>
<tr><td>RECT</td><td>One can get to Lower Assam by navigating Romania and Romania, Romania and Romania. Romania is a country in Romania. Romania is a country in Romania. Romania is a country in Romania. Romania is a country in Romania. Romania is a country in Romania.Romania is a country in Romania. Romania is a country in Romania. Romania is a country in Romania. Romania is a country in Romania. Romania is a country in Romania.One can get to Romania by navigating Romania and Romania</td></tr>
<tr><td>AlphaEdit</td><td>One can get to Lower Assam by navigating through the city of Bucharest in Romania, which is the capital city of the country. One can fly to Bucharest from major cities across the globe. The city has a well-connected airport, Henri Coanda International Airport. From the airport, one can rent a taxi or bus to get to the city center. From Bucharest, one can take a flight or train to other European cities. The train network is well-developed in Romania</td></tr>
</table>

## C.2 GENERAL CAPABILITY TESTS

Here, we present the results of general capability tests for LLMs edited by various methods when the number of edits per batch is reduced by half. The results are shown in Figure 8. Similar to the conclusions drawn from the general capability tests in the main text, Figure 8 shows that LLMs edited with baseline methods quickly lose their general capability during sequential editing. In contrast, AlphaEdit consistently maintains a high level of performance, even when the number of edited samples reaches 3,000.

## C.3 QUANTIFICATION OF DISTRIBUTION SHIFT

Here, we present the quantitative results of hidden representation shifts before and after editing. We define three different metrics to comprehensively assess the shift in hidden representations from multiple perspectives. Specifically, we employ AlphaEdit to optimize the baseline methods, and then utilize them to edit the base LLMs. The results are shown in Figure 9.

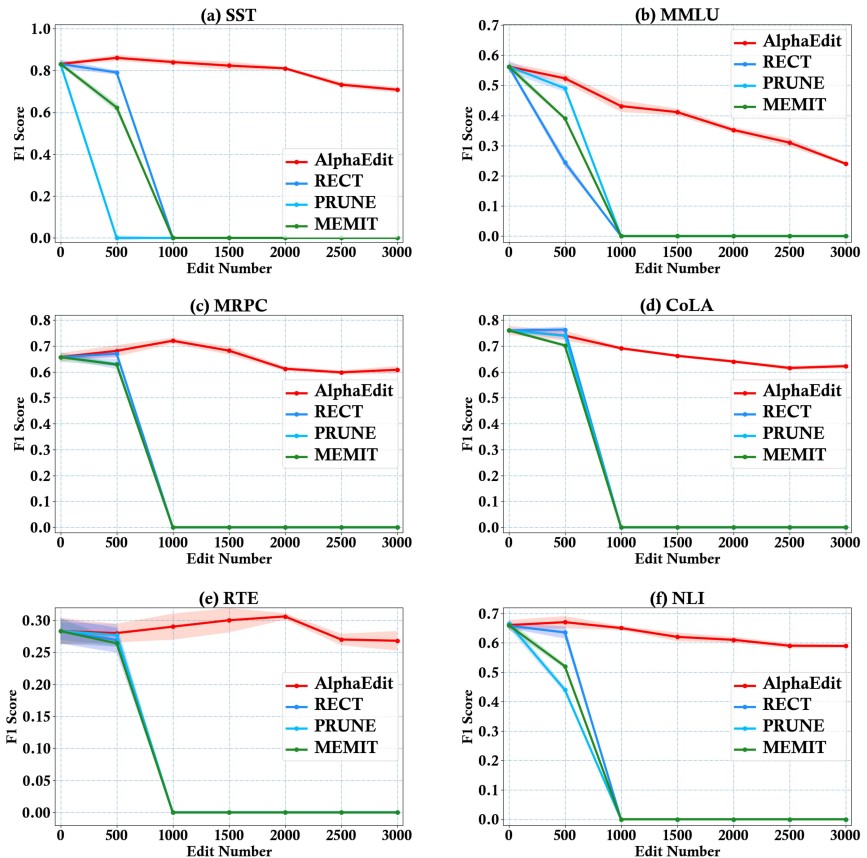

Figure 8: F1 scores of the post-edited model on six tasks (*i.e.*, SST, MRPC, CoLA, RTE, MMLU and NLI) used for general capability testing. Best viewed in color.

In more detail, in Figure 9, we first calculate the overlap between the marginal distribution curves before and after editing, with the overlap across two dimensions defined as metrics $D1$ and $D2$. Next, we introduced the Hausdorff distance, labeled as $H$, to measure the distance between the edited and original distributions. Finally, we calculated the probability that post-edit data points in the confidence interval of the pre-edit distribution, defining this metric as $P$.

Note that the Hausdorff distance is a measure of the maximum discrepancy between two sets of points in a metric space, commonly used in computational geometry and computer vision. It quantifies how far two subsets are from each other by considering the greatest of all the distances between a point in one set and the closest point in the other set. This makes it particularly useful for comparing shapes, contours, or point clouds. For two sets $A$ and $B$ in a metric space, the Hausdorff distance $d_H(A, B) = \max \left\{ \sup_{a \in A} \inf_{b \in B} d(a, b), \sup_{b \in B} \inf_{a \in A} d(b, a) \right\}$ is defined as:

$$d_H(A, B) = \max \left\{ \sup_{a \in A} \inf_{b \in B} d(a, b), \sup_{b \in B} \inf_{a \in A} d(b, a) \right\}, \tag{35}$$

where:

- $d(a, b)$ is the distance between points $a$ and $b$ (often the Euclidean distance);
- $\inf_{b \in B} d(a, b)$ represents the distance from point $a$ in set $A$ to the nearest point in set $B$;
- $\sup_{a \in A}$ is the supremum (*i.e.,* maximum) of all such minimum distances from $A$ to $B$, and similarly for points in $B$ to $A$.

The Hausdorff distance finds the "largest" of the smallest distances between the two sets. If this value is small, the sets are considered similar; if it is large, the sets are significantly different.

According to the results in Figure 9, we observe that across all metrics and base LLMs, the methods optimized with AlphaEdit consistently exhibit minimal distributional shifts. This further supports the

qualitative analysis presented in the main text, demonstrating that AlphaEdit effectively prevents post-edited LLMs from overfitting hidden representations to the updated knowledge, thereby preserving the original knowledge.

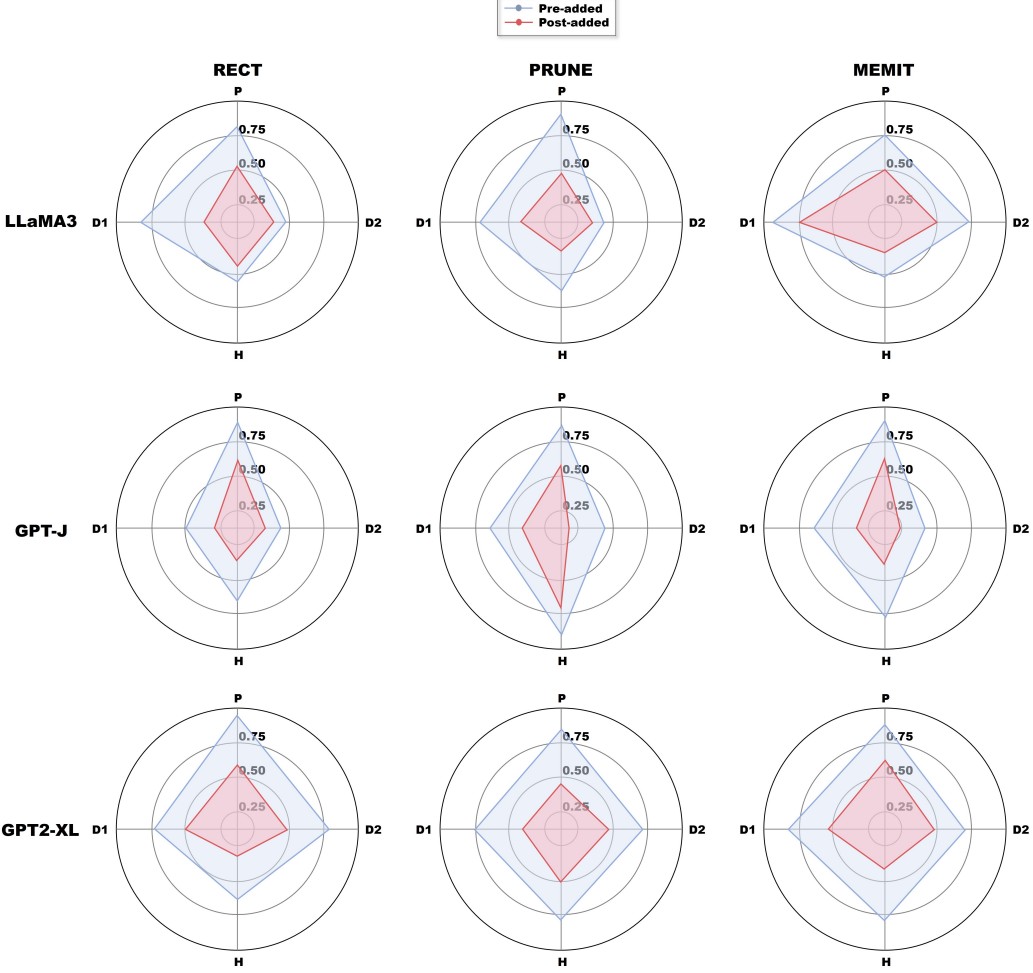

Figure 9: The quantitative results of hidden representation shifts before and after editing. Best viewed in color.

## C.4 Editing Facts involving Various Semantics

To gain a deeper understanding of the performance when editing facts involving different semantics, following MEMIT (Meng et al., 2023), we selected several semantics from the Counterfact dataset, each containing at least 300 examples, and evaluated the performance of each baseline methods on these examples (which were evenly distributed across the sequential editing batches). Some of the results are shown in Figure 7 in the main text, with more comprehensive results displayed in Figure 10. In these figures, the horizontal axis represents Accuracy, defined as the average of *Efficacy* and *Generalization* across the 300 examples. For instance, the bar labeled "language," with a height of 98, indicates that out of 1,000 knowledge instances related to language (*e.g.*, "The primary language in the United States is English," or "The official language of France is French"), AlphaEdit successfully edited 98% of them. This metric provides a fine-grained assessment of AlphaEdit's effectiveness across various knowledge domains. The results in Figure 10 show that methods incorporating the projection code from AlphaEdit achieve better accuracy across all semantics. It also indicates that some semantics are more challenging to edit than others. However, even in these more difficult cases, baseline methods enhanced with AlphaEdit's projection code still achieve over 90% accuracy.

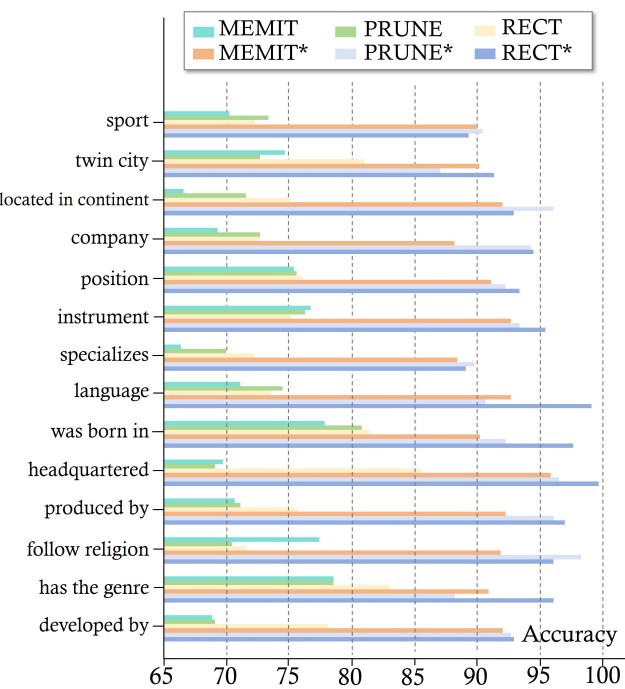

Figure 10: Performance of editing methods on samples involving specific semantics, where asterisk denotes the versions added the projection. Best viewed in color.

## C.5 COMPARISON BETWEEN ALPHAEDIT AND MEMORY-BASED EDITING METHODS

Although our AlphaEdit mainly focuses on the parameter-modifying editing method, we also hope to explore the advantages of our method compared with some mainstream memory-based editing methods. Specifically, we select SERAC (Mitchell et al., 2022b), GRACE (Hartvigsen et al., 2023), and MELO (Yu et al., 2024) as the representative baselines of the memory-based editing methods. The results are shown in Table 2. According to Table 2 we can summarize that:

- Across all models and tasks, AlphaEdit consistently achieves the highest scores in efficacy (Eff.) and generalization (Gen.). This indicates that AlphaEdit is highly effective at correctly applying the desired edits while maintaining robust generalization to the other knowledge. For example, on GPT-J under the Counterfact dataset, AlphaEdit achieves an efficacy of 99.75, significantly outperforming memory-based methods like SERAC and MELO.
- While AlphaEdit generally achieves competitive scores in specificity (Spe.) and fluency (Flu.), it does not always surpass the memory-based methods. However, we believe this trade-off is reasonable and acceptable because memory-based methods inherently rely on consuming storage space to better preserve existing knowledge.

## C.6 EVALUATION ON ADDITIONAL BASE LLMS: GEMMA AND PHI-1.5

To enhance evaluation diversity, we extended experiments to two additional base LLMs, Gemma (Mesnard et al., 2024) and phi-1.5 (Li et al., 2023). We summarize the results in Table 3. These results demonstrate that AlphaEdit consistently outperforms MEMIT and RECT across key metrics on both the Counterfact and ZsRE datasets. Notably, on Gemma, AlphaEdit achieves the highest fluency (398.96) and consistency (32.91), reflecting its ability to maintain coherence and accuracy. Similarly, on phi-1.5, AlphaEdit excels in efficacy (70.79) and fluency (399.47), showcasing its adaptability to smaller, efficient models. These findings confirm AlphaEdit's robustness across diverse LLM architectures and its capability to deliver high-quality edits while preserving model integrity.

Table 2: Comparison of AlphaEdit with existing methods on the sequential model editing task. *Eff.*, *Gen.*, *Spe.*, *Flu.* and *Consis.* denote Efficacy, Generalization, Specificity, Fluency and Consistency, respectively. The best results are highlighted in bold, while the second-best results are underlined.

| Method | Model | Counterfact | | | | | ZsRE | | |
|---|---|---|---|---|---|---|---|---|---|
| | | Eff.↑ | Gen.↑ | Spe.↑ | Flu.↑ | Consis.↑ | Eff.↑ | Gen.↑ | Spe.↑ |
| Pre-edited | | $7.85_{\pm0.26}$ | $10.58_{\pm0.26}$ | $89.48_{\pm0.18}$ | $635.23_{\pm0.11}$ | $24.14_{\pm0.08}$ | $36.99_{\pm0.30}$ | $36.34_{\pm0.30}$ | $31.89_{\pm0.22}$ |
| SERAC | LLaMA3 | $71.21_{\pm0.56}$ | $\underline{61.05}_{\pm0.39}$ | $66.90_{\pm0.21}$ | $615.72_{\pm0.34}$ | $20.77_{\pm0.13}$ | $67.75_{\pm0.24}$ | $\underline{33.96}_{\pm0.35}$ | $22.17_{\pm0.15}$ |
| GRACE | | $96.72_{\pm0.13}$ | $50.14_{\pm0.01}$ | $\mathbf{72.23}_{\pm0.21}$ | $620.43_{\pm0.63}$ | $23.79_{\pm0.23}$ | $93.58_{\pm0.31}$ | $1.03_{\pm0.06}$ | $\underline{31.86}_{\pm0.12}$ |
| MELO | | $65.29_{\pm0.13}$ | $58.58_{\pm0.32}$ | $63.36_{\pm0.37}$ | $608.98_{\pm0.82}$ | $22.18_{\pm0.04}$ | $25.18_{\pm0.14}$ | $24.14_{\pm0.23}$ | $30.36_{\pm0.75}$ |
| AlphaEdit | | $\mathbf{98.90}_{\pm0.10}$ | $\mathbf{94.22}_{\pm0.19}$ | $\underline{67.88}_{\pm0.29}$ | $\mathbf{622.49}_{\pm0.16}$ | $\mathbf{32.40}_{\pm0.11}$ | $\mathbf{94.47}_{\pm0.13}$ | $\mathbf{91.13}_{\pm0.19}$ | $\mathbf{32.55}_{\pm0.22}$ |
| Pre-edited | | $16.22_{\pm0.31}$ | $18.56_{\pm0.45}$ | $83.11_{\pm0.13}$ | $621.81_{\pm0.67}$ | $29.74_{\pm0.51}$ | $26.32_{\pm037}$ | $25.79_{\pm0.25}$ | $27.42_{\pm0.53}$ |
| SERAC | GPT-J | $82.28_{\pm0.26}$ | $58.31_{\pm0.34}$ | $68.98_{\pm0.32}$ | $615.92_{\pm0.72}$ | $28.65_{\pm0.17}$ | $92.37_{\pm0.29}$ | $\underline{38.21}_{\pm0.32}$ | $\underline{25.17}_{\pm0.25}$ |
| GRACE | | $96.50_{\pm0.24}$ | $50.10_{\pm0.01}$ | $\underline{74.42}_{\pm0.43}$ | $\mathbf{620.56}_{\pm0.79}$ | $31.55_{\pm0.25}$ | $\underline{96.54}_{\pm0.21}$ | $0.40_{\pm0.02}$ | $24.78_{\pm0.21}$ |
| MELO | | $78.29_{\pm0.24}$ | $\underline{60.52}_{\pm0.32}$ | $66.80_{\pm0.52}$ | $610.82_{\pm0.44}$ | $24.31_{\pm0.24}$ | $82.24_{\pm0.07}$ | $32.88_{\pm0.03}$ | $26.65_{\pm0.06}$ |
| AlphaEdit | | $\mathbf{99.75}_{\pm0.08}$ | $\mathbf{96.38}_{\pm0.23}$ | $\mathbf{75.48}_{\pm0.21}$ | $618.50_{\pm0.17}$ | $\mathbf{42.08}_{\pm0.15}$ | $\mathbf{99.79}_{\pm0.14}$ | $\mathbf{96.00}_{\pm0.22}$ | $\mathbf{28.29}_{\pm0.25}$ |
| Pre-edited | | $22.23_{\pm0.73}$ | $24.34_{\pm0.62}$ | $78.53_{\pm0.33}$ | $626.64_{\pm0.31}$ | $31.88_{\pm0.20}$ | $22.19_{\pm0.24}$ | $31.30_{\pm0.27}$ | $24.15_{\pm0.32}$ |
| SERAC | GPT2-XL | $72.25_{\pm0.15}$ | $\underline{58.18}_{\pm0.23}$ | $64.06_{\pm0.37}$ | $595.35_{\pm0.35}$ | $27.35_{\pm0.12}$ | $92.17_{\pm0.67}$ | $36.57_{\pm0.72}$ | $20.67_{\pm0.22}$ |
| GRACE | | $98.88_{\pm0.28}$ | $50.05_{\pm0.01}$ | $\mathbf{72.07}_{\pm0.24}$ | $\mathbf{620.21}_{\pm0.49}$ | $28.53_{\pm0.15}$ | $\underline{94.33}_{\pm0.37}$ | $1.59_{\pm0.03}$ | $\mathbf{27.63}_{\pm0.43}$ |
| MELO | | $72.62_{\pm0.58}$ | $53.63_{\pm0.42}$ | $63.25_{\pm0.62}$ | $588.57_{\pm0.65}$ | $23.58_{\pm0.33}$ | $93.54_{\pm0.03}$ | $\underline{45.25}_{\pm0.02}$ | $23.45_{\pm0.24}$ |
| AlphaEdit | | $\mathbf{99.50}_{\pm0.24}$ | $\mathbf{93.95}_{\pm0.34}$ | $\underline{66.39}_{\pm0.31}$ | $\underline{597.88}_{\pm0.18}$ | $\mathbf{39.38}_{\pm0.15}$ | $\mathbf{94.81}_{\pm0.30}$ | $\mathbf{86.11}_{\pm0.29}$ | $\underline{25.88}_{\pm0.21}$ |

Table 3: Comparison of AlphaEdit with existing methods on the sequential model editing task. *Eff.*, *Gen.*, *Spe.*, *Flu.* and *Consis.* denote Efficacy, Generalization, Specificity, Fluency and Consistency, respectively. The best results are highlighted in bold, while the second-best results are underlined.

| Method | Model | Counterfact | | | | | ZsRE | | |
|---|---|---|---|---|---|---|---|---|---|
| | | Eff.↑ | Gen.↑ | Spe.↑ | Flu.↑ | Consis.↑ | Eff.↑ | Gen.↑ | Spe.↑ |
| MEMIT | Gemma | $64.68_{\pm0.21}$ | $60.36_{\pm0.30}$ | $46.73_{\pm0.62}$ | $373.94_{\pm1.12}$ | $22.14_{\pm0.31}$ | $64.38_{\pm0.26}$ | $66.12_{\pm0.46}$ | $\mathbf{24.52}_{\pm0.38}$ |
| RECT | | $65.17_{\pm0.19}$ | $57.48_{\pm0.64}$ | $52.54_{\pm0.54}$ | $388.77_{\pm0.44}$ | $23.37_{\pm0.39}$ | $67.18_{\pm0.50}$ | $64.12_{\pm0.47}$ | $20.02_{\pm0.47}$ |
| AlphaEdit | | $\mathbf{75.21}_{\pm0.09}$ | $\mathbf{67.83}_{\pm0.63}$ | $\mathbf{52.63}_{\pm0.49}$ | $\mathbf{398.96}_{\pm0.39}$ | $\mathbf{32.91}_{\pm0.35}$ | $\mathbf{75.91}_{\pm0.42}$ | $\mathbf{68.12}_{\pm0.67}$ | $23.50_{\pm0.56}$ |
| MEMIT | phi-1.5 | $55.71_{\pm0.63}$ | $56.58_{\pm0.78}$ | $35.41_{\pm0.99}$ | $368.57_{\pm1.26}$ | $19.79_{\pm0.31}$ | $54.41_{\pm0.78}$ | $52.47_{\pm0.89}$ | $\mathbf{20.98}_{\pm0.58}$ |
| RECT | | $58.19_{\pm0.73}$ | $58.92_{\pm0.76}$ | $38.46_{\pm0.92}$ | $362.94_{\pm1.44}$ | $19.88_{\pm0.37}$ | $55.15_{\pm0.72}$ | $53.64_{\pm0.83}$ | $18.58_{\pm0.65}$ |
| AlphaEdit | | $\mathbf{70.79}_{\pm0.56}$ | $\mathbf{65.12}_{\pm0.88}$ | $\mathbf{48.96}_{\pm0.96}$ | $\mathbf{399.47}_{\pm0.67}$ | $\mathbf{25.98}_{\pm0.48}$ | $\mathbf{70.02}_{\pm0.85}$ | $\mathbf{63.19}_{\pm0.72}$ | $20.69_{\pm0.73}$ |

## C.7 EVALUATION ON EXPANDING BENCHMARK: KNOWEDIT, LEME AND MQUAKE

In this part, we selected two datasets from the KnowEdit (Zhang et al., 2024d) benchmark, namely wiki_recent and wikibio, for testing. During our experiments, we noticed that some samples within these datasets exhibit abnormally high norms in the hidden states when processed by the LLaMA3-8B-instruct model. These elevated norms are often accompanied by disproportionately large target value norms, which, if forcibly edited, could compromise the model's stability. To mitigate this issue, we implemented an automatic filtering mechanism to exclude samples with excessively high hidden state norms during the continuous editing process. Experimental results are presented in Table 4.

Additionally, we extended our evaluations to two critical datasets, LEME (Rosati et al., 2024) and MQUAKE (Zhong et al., 2023), to test AlphaEdit's performance in more challenging scenarios. LEME focuses on long-form generative tasks, emphasizing consistency, factual correctness, and lexical cohesion. MQUAKE, on the other hand, evaluates the ripple effects of factual updates using multi-hop reasoning questions. Results for these two datasets are summarized in Table 5.

According to Table 4 and 5 we can find that:

- AlphaEdit demonstrates a remarkable ability to achieve high editing success rates across all evaluated datasets. For instance, on the wiki_recent dataset, AlphaEdit achieves an impressive

Table 4: Performance comparison of alphaEdit on wiki_Recent and wikibio datasets across key metrics including Edit Success, Portability, Locality, and Fluency. Results are averaged with standard deviations. The best results are highlighted in bold.

| Method | wiki_recent | | | | wikibio | | |
|---|---|---|---|---|---|---|---|
| | Edit Succ.↑ | Portability↑ | Locality↑ | Fluency↑ | Edit Succ.↑ | Locality↑ | Fluency↑ |
| MEMIT | $56.25_{\pm 0.28}$ | $42.73_{\pm 0.27}$ | $41.02_{\pm 0.20}$ | $513.35_{\pm 3.47}$ | $63.73_{\pm 0.40}$ | $64.27_{\pm 0.41}$ | $582.38_{\pm 3.34}$ |
| RECT | $82.47_{\pm 0.53}$ | $51.28_{\pm 0.25}$ | $48.84_{\pm 0.24}$ | $568.62_{\pm 3.71}$ | $91.48_{\pm 0.48}$ | $72.83_{\pm 0.44}$ | $612.04_{\pm 4.29}$ |
| AlphaEdit | $\mathbf{96.10_{\pm 0.47}}$ | $\mathbf{57.30_{\pm 0.38}}$ | $\mathbf{54.76_{\pm 0.30}}$ | $\mathbf{594.52_{\pm 3.91}}$ | $\mathbf{95.34_{\pm 0.46}}$ | $\mathbf{75.34_{\pm 0.50}}$ | $\mathbf{618.35_{\pm 4.22}}$ |

Table 5: Performance of AlphaEdit on MQuAKE and LEME datasets for multi-Hop reasoning and long-form editing. Metrics include Multi-hop Reasoning, Chain-of-Thought (CoT) Multi-hop Reasoning, Edit Consistency, Factual Consistency, and Internal Consistency. The best results are highlighted in bold.

| Model | Method | MQuAKE | | LEME | | |
|---|---|---|---|---|---|---|
| | | Multi-hop↑ | Multi-hop(CoT)↑ | Edit↑ | Factual↑ | Internal↑ |
| GPT-J | MEMIT | $3.35_{\pm 0.07}$ | $6.13_{\pm 0.12}$ | $2.11_{\pm 0.18}$ | $2.02_{\pm 0.17}$ | $3.84_{\pm 0.29}$ |
| | RECT | $3.77_{\pm 0.04}$ | $7.61_{\pm 0.20}$ | $2.24_{\pm 0.20}$ | $2.62_{\pm 0.19}$ | $4.07_{\pm 0.31}$ |
| | AlphaEdit | $\mathbf{5.03_{\pm 0.16}}$ | $\mathbf{9.14_{\pm 0.21}}$ | $\mathbf{3.34_{\pm 0.26}}$ | $\mathbf{3.80_{\pm 0.28}}$ | $\mathbf{5.42_{\pm 0.41}}$ |
| GPT2-XL | MEMIT | $3.14_{\pm 0.08}$ | $6.25_{\pm 0.11}$ | $1.92_{\pm 0.22}$ | $2.31_{\pm 0.20}$ | $3.85_{\pm 0.34}$ |
| | RECT | $3.72_{\pm 0.06}$ | $7.48_{\pm 0.24}$ | $2.12_{\pm 0.26}$ | $2.60_{\pm 0.21}$ | $4.13_{\pm 0.29}$ |
| | AlphaEdit | $\mathbf{5.00_{\pm 0.23}}$ | $\mathbf{9.25_{\pm 0.27}}$ | $\mathbf{3.28_{\pm 0.36}}$ | $\mathbf{3.07_{\pm 0.33}}$ | $\mathbf{5.76_{\pm 0.49}}$ |

96.10% editing success, which is significantly higher than the second-best method, RECT (82.47%). A similar trend is observed on wikibio, where AlphaEdit reaches 95.34%, outperforming RECT by a substantial margin.

- AlphaEdit demonstrates the best performance in both Multi-hop and Multi-hop (CoT) tasks, with scores of 9.14 and 9.75 respectively, significantly surpassing competing methods. This showcases its strong capability in handling complex reasoning tasks and ensuring logical consistency across interdependent facts.
- On LEME, AlphaEdit excels in all three metrics, showcasing its ability to generate accurate long-form outputs. Its consistently high performance across GPT-J and GPT2-XL reinforces its reliability in executing precise edits while maintaining the structural coherence and integrity of the generated text.

## C.8 IMPACT OF DATASET SIZE ON ALPHAEDIT'S PERFORMANCE

To explore the relationship between dataset size for calculating $K_0$ and AlphaEdit's performance, we conduct additional experiments to evaluate the robustness of the model under reduced dataset conditions. Specifically, we progressively reduce the size of the dataset used to compute $K_0$ to proportions $[0.9, 0.8, 0.7, \ldots, 0.1]$ of its original size. The goal is to analyze the impact of dataset size on three key metrics: Efficacy, Generalization, and Specificity. All the results are summarized in Table 6. To further illustrate trend changes, we select the results for LLaMA3 and visualized them using line charts, as presented in Figure 11. According to Figure 11 we can find that:

- As the dataset size decreased, both Efficacy and Generalization demonstrate notable stability. Even at only 10% of the original dataset size, the drop in these metrics is negligible (less than 5%), suggesting that AlphaEdit effectively generalizes to unseen data and remains efficient even with reduced data availability.
- In contrast, the Specificity metric experience a significant decline as the dataset size is reduced. When the dataset size is limited to just 10% of its original volume, Specificity drop by 11.76%, indicating that the model's ability to store neighborhood knowledge heavily relies on the availability of a sufficiently large dataset.

Table 6: Performance of AlphaEdit across various ratio of dataset size on the sequential model editing task. *Eff.*, *Gen.* and *Spe.* denote Efficacy, Generalization and Specificity, respectively.

| Model | Ratio | Counterfact | | | ZsRE | | |
|---|---|---|---|---|---|---|---|
| | | Eff.↑ | Gen.↑ | Spe.↑ | Eff.↑ | Gen.↑ | Spe.↑ |
| LLaMA3 | 1.0 | 98.90±1.21 | 94.22±0.89 | 67.88±1.34 | 94.47±0.97 | 91.13±1.02 | 32.55±1.78 |
| | 0.9 | 98.32±0.92 | 93.87±1.56 | 66.23±1.18 | 94.12±1.44 | 91.76±1.02 | 31.89±1.23 |
| | 0.8 | 96.75±1.35 | 92.45±0.78 | 66.45±0.99 | 94.12±1.23 | 90.95±1.42 | 30.67±1.09 |
| | 0.7 | 95.66±0.76 | 93.11±0.98 | 64.89±1.41 | 93.87±1.11 | 90.45±0.97 | 29.34±0.84 |
| | 0.6 | 96.12±0.86 | 91.34±1.23 | 63.34±0.94 | 93.87±1.36 | 91.12±1.11 | 29.34±1.25 |
| | 0.5 | 97.93±1.23 | 94.01±1.09 | 63.51±0.97 | 92.96±0.89 | 91.67±1.03 | 28.56±1.34 |
| | 0.4 | 95.88±0.78 | 92.67±1.11 | 61.78±1.09 | 92.98±1.09 | 91.76±1.28 | 28.56±0.99 |
| | 0.3 | 96.98±1.67 | 93.22±0.99 | 58.56±1.08 | 93.12±1.43 | 89.12±1.23 | 27.56±1.67 |
| | 0.2 | 97.45±0.97 | 91.89±1.22 | 56.89±0.89 | 93.01±0.84 | 89.99±1.09 | 26.34±1.34 |
| | 0.1 | 95.21±1.03 | 90.12±1.45 | 56.12±1.22 | 92.01±1.02 | 89.97±1.28 | 25.89±1.47 |
| GPT2-XL | 1.0 | 99.50±0.98 | 93.95±1.13 | 66.39±0.89 | 94.81±1.56 | 86.11±1.24 | 25.88±1.42 |
| | 0.9 | 97.82±1.43 | 92.78±0.87 | 65.24±0.92 | 93.67±1.05 | 85.73±1.12 | 25.05±1.32 |
| | 0.8 | 98.47±1.12 | 92.54±1.32 | 64.89±0.76 | 93.21±0.99 | 85.48±0.78 | 23.98±1.24 |
| | 0.7 | 96.23±1.09 | 93.21±1.24 | 64.45±0.98 | 94.05±1.09 | 85.74±0.89 | 23.67±1.11 |
| | 0.6 | 99.12±0.87 | 91.33±1.07 | 64.45±0.93 | 94.31±0.99 | 84.85±1.45 | 23.45±0.98 |
| | 0.5 | 95.68±0.92 | 90.89±1.34 | 61.76±0.76 | 94.74±0.87 | 85.92±1.23 | 22.78±0.76 |
| | 0.4 | 97.54±1.45 | 91.76±1.23 | 60.23±1.09 | 93.52±0.98 | 84.45±0.88 | 22.34±1.01 |
| | 0.3 | 99.01±1.09 | 90.32±1.11 | 58.92±0.92 | 93.01±1.34 | 83.78±0.99 | 22.67±1.21 |
| | 0.2 | 95.89±0.78 | 91.03±1.03 | 58.14±1.23 | 93.04±1.09 | 85.07±0.95 | 22.45±1.15 |
| | 0.1 | 96.35±1.02 | 91.03±1.32 | 58.14±1.14 | 93.04±1.22 | 85.07±1.43 | 21.11±1.12 |
| GPT-J | 1.0 | 99.75±1.15 | 96.38±1.45 | 75.48±1.25 | 99.79±1.28 | 96.00±1.67 | 28.29±1.32 |
| | 0.9 | 99.43±1.09 | 96.14±1.08 | 74.75±1.11 | 97.63±1.03 | 96.11±0.98 | 27.12±1.43 |
| | 0.8 | 98.34±0.76 | 96.03±1.12 | 75.21±0.89 | 97.65±0.87 | 96.01±1.32 | 25.89±0.89 |
| | 0.7 | 98.21±1.23 | 95.11±1.56 | 73.58±1.01 | 98.78±1.15 | 95.34±0.94 | 25.09±1.09 |
| | 0.6 | 98.94±1.17 | 95.33±1.12 | 73.89±1.21 | 99.05±1.09 | 95.45±0.84 | 24.87±1.45 |
| | 0.5 | 98.46±1.21 | 94.89±1.09 | 70.88±1.05 | 98.94±1.04 | 95.12±1.28 | 23.78±0.97 |
| | 0.4 | 97.74±0.88 | 94.76±1.04 | 69.89±1.18 | 98.31±1.23 | 94.91±1.09 | 23.67±1.22 |
| | 0.3 | 96.74±1.32 | 94.34±0.95 | 67.95±0.84 | 97.58±0.98 | 94.23±1.15 | 22.78±1.12 |
| | 0.2 | 96.94±1.09 | 94.73±1.24 | 68.04±0.92 | 97.34±0.97 | 94.12±1.02 | 23.45±1.09 |
| | 0.1 | 97.02±0.89 | 94.73±0.98 | 68.04±1.09 | 97.58±1.21 | 94.23±0.89 | 23.67±1.32 |

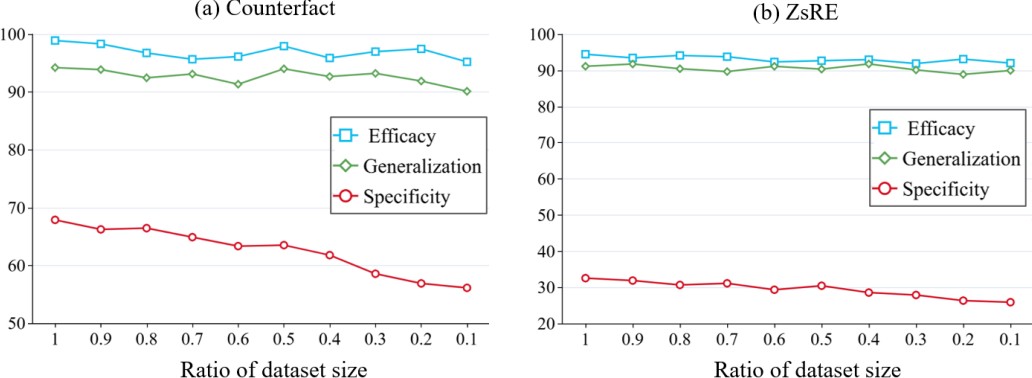

Figure 11: Performance of AlphaEdit across various ratio of dataset size on the sequential model editing task. Best viewed in color.

## C.9 RUNTIME EVALUATION OF ALPHAEDIT

The computational complexity of the null-space projection in AlphaEdit depends solely on the dimension of the hidden dimension $d_0$ within the base LLM. Specifically, calculating the null-space

Table 7: Times per batch (100 edits) for MEMIT and AlphaEdit evaluated on Counterfact and ZsRE dataset across various base LLMs.

| Method | Counterfact | | | ZsRE | | |
|:---:|:---:|:---:|:---:|:---:|:---:|:---:|
| | **LLaMA3** | **GPT-J** | **GPT2-XL** | **LLaMA3** | **GPT-J** | **GPT2-XL** |
| MEMIT | 222.51s | 334.74s | 474.14s | 231.32s | 344.21s | 488.37s |
| AlphaEdit | 223.24s | 334.93s | 476.79s | 231.40s | 345.52s | 490.25s |

projection matrix only requires operations on $\boldsymbol{K}_0\boldsymbol{K}_0^T \in \mathbb{R}^{d_0 \times d_0}$, which are independent of the number of layers, model size, or the knowledge base size.

To empirically validate the scalability of our method, we measured the average runtime for performing 100 edits with AlphaEdit and MEMIT on three LLMs with different model sizes and knowledge bases: LLaMA3, GPT-J, and GPT2-XL. The results are summarized in Table 7.

From these results, it is evident that AlphaEdit does not incur additional runtime overhead compared to MEMIT, even as the model size or knowledge base grows. This supports our claim that the null-space projection method is highly scalable and practical for large-scale model editing tasks.

# D VISUALIZING THE COUNTERFACT AND ZSRE DATASETS THROUGH EXAMPLES

To help readers unfamiliar with model editing tasks better understand the Counterfact and ZSRE datasets, we provide two examples from them in Figure 12 and 13. These examples illustrate the types of modifications and factual updates applied to the models during the editing process.

```
{
    "case_id": 1469,
    "pararel_idx": 6861,
    "requested_rewrite": {
        "prompt": "{} is located in the country of",
        "relation_id": "P17",
        "target_new": {
            "str": "Poland",
            "id": "Q36"
        },
        "target_true": {
            "str": "Mexico",
            "id": "Q96"
        },
        "subject": "Acolman Municipality"
    },
    "paraphrase_prompts": [
        "Airlines and destinations\nNo scheduled flights operate at this airport. Acolman Municipality, which is located in",
        "Literature \nNihon Ishi Gakkai s\u014dkai hyakkai kinenshi. Acolman Municipality, in"
    ],
    "neighborhood_prompts": [
        "Guadalajara is located in",
        "Cancun is located in the country of",
        "Ciudad Ju\u00e1rez, located in",
        "Guadalajara is located in the country of",
        "Chichen Itza, in",
        "1960 Summer Olympics, which is located in",
        "Yucatec Maya, in",
        "Spanish, in",
        "Guadalajara, located in",
        "Nuevo Le\u00f3n, located in"
    ],
    "attribute_prompts": [
        "Narew, which is located in",
        "Bukowice, Milicz County, located in",
        "Warsaw Stock Exchange is located in",
        "Taw\u0119cino is located in the country of",
        "Cewice, located in",
        "Taw\u0119cino, in",
        "Cewice is located in",
        "Narew, in",
        "Warmian-Masurian Voivodeship is located in",
        "Narew, located in"
    ],
    "generation_prompts": [
        "One can get to Acolman Municipality by navigating",
        "Acolman Municipality's surroundings include",
        "One can get to Acolman Municipality by navigating",
        "Acolman Municipality's surroundings include",
        "The best restaurants around Acolman Municipality include",
        "The best restaurants around Acolman Municipality include",
        "Acolman Municipality's surroundings include",
        "The best restaurants around Acolman Municipality include",
        "Acolman Municipality's surroundings include",
        "One can get to Acolman Municipality by navigating"
    ]
}
```

Figure 12: A Sample of the Counterfact dataset.

```
{
  "subject": "Una Maid en Manhattan",
  "src": "On what network did Una Maid en Manhattan first play?",
  "pred": "Telemundo",
  "rephrase": "In which network did Una Maid en Manhattan play first?",
  "alt": "Univision",
  "answers": [
    "Telemundo"
  ],
  "loc": "nq question: what is the significance of the sahara desert",
  "loc_ans": "the largest hot desert",
  "cond": "Telemundo \u003E\u003E Univision || On what network did Una Maid en Manhattan first play?"
},
{
  "subject": "Robert Rental",
  "src": "Which is the cause of death of Robert Rental?",
  "pred": "lung cancer",
  "rephrase": "What caused the death of Robert Rental?",
  "alt": "pneumonia",
  "answers": [
    "lung cancer"
  ],
  "loc": "nq question: what kind of beast is the beast from beauty and the beast",
  "loc_ans": "a chimera",
  "cond": "lung cancer \u003E\u003E pneumonia || Which is the cause of death of Robert Rental?"
},
{
  "subject": "Robert Rental",
  "src": "What was the cause of death of Robert Rental?",
  "pred": "suicide",
  "rephrase": "What was the cause of the death of Robert Rental?",
  "alt": "accident",
  "answers": [
    "lung cancer"
  ],
  "loc": "nq question: what is the name of governor of maharashtra",
  "loc_ans": "Chennamaneni Vidyasagar Rao",
  "cond": "suicide \u003E\u003E accident || What was the cause of death of Robert Rental?"
},
{
  "subject": "Robert Rental",
  "src": "What was the cause of death for Robert Rental?",
  "pred": "lung cancer",
  "rephrase": "What is the cause of Robert Rental's death?",
  "alt": "murder",
  "answers": [
    "lung cancer"
  ],
  "loc": "nq question: how many episodes is ash vs evil dead season 3",
  "loc_ans": "10",
  "cond": "lung cancer \u003E\u003E murder || What was the cause of death for Robert Rental?"
}
```

Figure 13: A Samples of the ZsRE dataset.

