# OpenReview forum: "AlphaEdit: Null-Space Constrained Knowledge Editing for Language Models"
_ICLR.cc/2025/Conference — ICLR 2025 Oral_

### Official Review · Reviewer_cdNJ · 2024-11-04

**Soundness:** 4
**Presentation:** 4
**Contribution:** 4
**Rating:** 8
**Confidence:** 4

**Summary:**

Post training an LLM can often cause it to disrupt its originally preserved knowledge. To circumvent this, AlphaEdit utilizes Null Space Projection to preserve old knowledge while injecting the new knowledge effectively. Results show that AlphaEdit significantly reduces the domain shift between pre and post edits compared to existing methods.

**Strengths:**

1. The paper is well-written and easy to follow. The authors explained the null space and how to leverage null space projection to optimize the model editing objective well.
2. I think the choice of RQs is well thought out and thorough as well. The paper answered most of the questions I had about AlphaEdit.
3. I think figure 6 is interesting to show how AlphaEdit can generalize to existing methods.

**Weaknesses:**

1. The paper did not mention the correlation between the accuracy and the dataset size. More concretely, how much data is needed for AlphaEdit to work well?

**Questions:**

1. In line 174, ‘B is in the null space of B’ -> change to ‘B is in the null space of A’
2. For figure 7(a), what does the ylabel refer to?
3. Minor presentation suggestion: In figure 5, the pre and post edited distributions are difficult to set apart due to color choice. Picking two contrasting colors similar to Memit would be great for the final version.

---

> ### Author Response · Authors · 2024-11-19
> **Response to Reviewer cdNJ (1/2)**
>
> Dear Reviewer cdNJ:
>
>
>
> Thank you for your kind words and positive feedback of our novelty, presentation and effectiveness! Your approval is the great encouragement for us and motivates us to continue advancing our work.
>
>
>
>
> Below, we meticulously provide responses to each of your comments and outline the modifications made to the manuscript. All revisions are highlighted in blue.
>
>
> --------------------------
>
> ## *W1: The paper did not mention the correlation between the accuracy and the dataset size.*
>
> Thank you for raising this important concern. We acknowledge that the paper did not address the correlation between dataset size and AlphaEdit's performance.
>
>
> Following your suggestion, we conducted additional experiments by reducing the size of the dataset used to compute $K_0$  to proportions [0.9, 0.8, 0.7, ... , 0.1] of its original size and observed the impact on AlphaEdit's performance. Detailed results and analyses are provided in `Appendix C.8` (Page 26-27, Line 1403-1475).
>
>
> For your convenience, we summarize the key results and observations here:
>
>
> |||||||||
> |:-:|:-:|:-:|:-:|:-:|:-:|:-:|:-:|
> ||||$\text{Counterfact}$|||$\text{ZsRE}$||
> |$\text{Model}$|$\text{Ratio}$|$\text{Eff.}$$\uparrow$|$\text{Gen.}$$\uparrow$|$\text{Spe.}$$\uparrow$|$\text{Eff.}$$\uparrow$|$\text{Gen.}$$\uparrow$|$\text{Spe.}$$\uparrow$|
> | |1.0|$98.90±1.21$|$94.22±0.89$|$67.88±1.34$|$94.47±0.97$|$91.13±1.02$|$32.55±1.78$|
> ||0.9|$98.32±0.92$|$93.87±1.56$|$66.23±1.18$|$94.12±1.44$|$91.76±1.02$|$31.89±1.23$|
> ||0.8|$96.75±1.35$|$92.45±0.78$|$66.45±0.99$|$94.12±1.23$|$90.95±1.42$|$30.67±1.09$|
> | |0.6|$96.12±0.86$|$91.34±1.23$|$63.34±0.94$|$93.87±1.36$|$91.12±1.11$|$29.34±1.25$|
> |$\text{LLaMA3}$|0.5|$97.93±1.23$|$94.01±1.09$|$63.51±0.97$|$92.96±0.89$|$91.67±1.03$|$28.56±1.34$|
> ||0.4|$95.88±0.78$|$92.67±1.11$|$61.78±1.09$|$92.98±1.09$|$91.76±1.28$|$28.56±0.99$|
> ||0.3|$96.98±1.67$|$93.22±0.99$|$58.56±1.08$|$93.12±1.43$|$89.12±1.23$|$27.56±1.67$|
> ||0.2|$97.45±0.97$|$91.89±1.22$|$56.89±0.89$|$93.01±0.84$|$89.99±1.09$|$26.34±1.34$|
> ||0.1|$95.21±1.03$|$90.12±1.45$|$56.12±1.22$|$92.01±1.02$|$89.97±1.28$|$25.89±1.47$|
> |||||||||
>
>
> |||||||||
> |:-:|:-:|:-:|:-:|:-:|:-:|:-:|:-:|
> ||||$\text{Counterfact}$|||$\text{ZsRE}$||
> |$\text{Model}$|$\text{Ratio}$|$\text{Eff.}$$\uparrow$|$\text{Gen.}$$\uparrow$|$\text{Spe.}$$\uparrow$|$\text{Eff.}$$\uparrow$|$\text{Gen.}$$\uparrow$|$\text{Spe.}$$\uparrow$|
> | |1.0|$99.50±0.98$|$93.95±1.13$|$66.39±0.89$|$94.81±1.56$|$86.11±1.24$|$25.88±1.42$|
> ||0.9|$97.82±1.43$|$92.78±0.87$|$65.24±0.92$|$93.67±1.05$|$85.73±1.12$|$25.05±1.32$|
> ||0.8|$98.47±1.12$|$92.54±1.32$|$64.89±0.76$|$93.21±0.99$|$85.48±0.78$|$23.98±1.24$|
> ||0.6|$99.12±0.87$|$91.33±1.07$|$64.45±0.93$|$94.31±0.99$|$84.85±1.45$|$23.45±0.98$|
> |$\text{GPT2-XL}$|0.5|$95.68±0.92$|$90.89±1.34$|$61.76±0.76$|$94.74±0.87$|$85.92±1.23$|$22.78±0.76$|
> ||0.4|$97.54±1.45$|$91.76±1.23$|$60.23±1.09$|$93.52±0.98$|$84.45±0.88$|$22.34±1.01$|
> ||0.3|$99.01±1.09$|$90.32±1.11$|$58.92±0.92$|$93.01±1.34$|$83.78±0.99$|$22.67±1.21$|
> ||0.2|$95.89±0.78$|$91.03±1.03$|$58.14±1.23$|$93.04±1.09$|$85.07±0.95$|$22.45±1.15$|
> ||0.1|$96.35±1.02$|$91.03±1.32$|$58.14±1.14$|$93.04±1.22$|$85.07±1.43$|$21.11±1.12$|
> |||||||||

---

> ### Author Response · Authors · 2024-11-19
> **Response to Reviewer cdNJ (2/2)**
>
> |||||||||
> |:-:|:-:|:-:|:-:|:-:|:-:|:-:|:-:|
> ||||$\text{Counterfact}$|||$\text{ZsRE}$||
> |$\text{Model}$|$\text{Ratio}$|$\text{Eff.}$$\uparrow$|$\text{Gen.}$$\uparrow$|$\text{Spe.}$$\uparrow$|$\text{Eff.}$$\uparrow$|$\text{Gen.}$$\uparrow$|$\text{Spe.}$$\uparrow$|
> | |1.0|$99.75±1.15$|$96.38±1.45$|$75.48±1.25$|$99.79±1.28$|$96.00±1.67$|$28.29±1.32$|
> ||0.9|$99.43±1.09$|$96.14±1.08$|$74.75±1.11$|$97.63±1.03$|$96.11±0.98$|$27.12±1.43$|
> ||0.8|$98.34±0.76$|$96.03±1.12$|$75.21±0.89$|$97.65±0.87$|$96.01±1.32$|$25.89±0.89$|
> ||0.6|$98.94±1.17$|$95.33±1.12$|$73.89±1.21$|$99.05±1.09$|$95.45±0.84$|$24.87±1.45$|
> |$\text{GPT-J}$|0.5|$98.46±1.21$|$94.89±1.09$|$70.88±1.05$|$98.94±1.04$|$95.12±1.28$|$23.78±0.97$|
> ||0.4|$97.74±0.88$|$94.76±1.04$|$69.89±1.18$|$98.31±1.23$|$94.91±1.09$|$23.67±1.22$|
> ||0.3|$96.74±1.32$|$94.34±0.95$|$67.95±0.84$|$97.58±0.98$|$94.23±1.15$|$22.78±1.12$|
> ||0.2|$96.94±1.09$|$94.73±1.24$|$68.04±0.92$|$97.34±0.97$|$94.12±1.02$|$23.45±1.09$|
> ||0.1|$97.02±0.89$|$94.73±0.98$|$68.04±1.09$|$97.58±1.21$|$94.23±0.89$|$23.67±1.32$|
> |||||||||
>
>
> According to the above tables, we can find that:
>
>  - **As the dataset size decreased, both Efficacy and Generalization demonstrate notable stability.** Even at only 10\% of the original dataset size, the drop in these metrics is negligible (less than 5\%), suggesting that AlphaEdit effectively generalizes to unseen data and remains efficient even with reduced data availability.
>
>  - **In contrast, the Specificity metric experience a significant decline as the dataset size is reduced.** When the dataset size is limited to just 10\% of its original volume, Specificity drop by 11.76\%, indicating that the model's ability to store neighborhood knowledge heavily relies on the availability of a sufficiently large dataset.
>
>
> Due to time constraints during the discussion phase, we have completed these experiments with LLaMA3, GPT-J and GPT2-XL as the base model on the Counterfact and ZsRE datasets. Additional tests with other base models (*e.g.*, Gemma, Phi) and datasets (*e.g.*, LongformEvaluation, MQUAKE, KnowEdit) are already underway. **We will update the reversion as soon as we complete these experiments.**
>
>
>
> Hope our additional experiments could address your concerns!
>
>
> --------------------------
>
>
> ## *Q1 & Q2: In line 174, change $B$ to $A$. For figure 7(a), what does the ylabel refer to?*
>
> Thanks for your comments. Based on your suggestions, we have made the following revisions in the updated manuscript:
>
>
>  - **Line 174**：Replaced "B" with the correct symbol "A": B is in the null space of A if and only if BA=0.
>
>
>  - **Line 464**：Added clarification for the vertical and horizontal axis in Figure 7 (a): The vertical and horizontal axis represent the categories of knowledge and the accuracy of LLM responses involving this knowledge, respectively.
>
>
> --------------------------
>
>
> ## *Q3:  Minor presentation suggestion: Picking contrasting colors would be great in Figure 5.*
>
>
> Thank you for your valuable suggestion! We have provided an updated version of the figure with adjusted colors in `Lines 1943–1511` of the revised manuscript (enlarged for your convenience to facilitate comparison).
>
>
> If you feel this meets your expectations, we will apply the same adjustments to all scatter plots in the camera-ready version.  Should you find further refinements necessary, please let us know—we would be happy to make additional adjustments.
>
>
>
>  Hope that these updates could meet your expectations, and **we would be thrilled if you could let us know whether your concerns have been addressed or if you have any follow-up questions!**
>
>
> --------------------------
>
>
>
> Once again, we deeply appreciate your thoughtful and encouraging feedback. Your suggestions have not only enhanced the current work but have also inspired us to to **keep moving forward and contributing to the community**!
>
>
>
>
> Best,
>
>
>
>
> Authors of Paper 3792

---

> ### Author Response · Authors · 2024-11-25
> **Heartfelt Gratitude to Reviewer cdNJ**
>
> Dear Reviewer cdNJ,
>
> We would like to extend our heartfelt gratitude for your thoughtful and constructive suggestions on our manuscript. Your insightful feedback has significantly strengthened the overall quality of our paper.
>
>
> We hope that our responses have effectively clarified your concerns and provided satisfactory explanations. **If there are any remaining questions or additional points you would like to discuss, we would be more than happy to engage in further dialogue to address them.**
>
>
> Once again, we sincerely appreciate the time and effort you have devoted to reviewing our manuscript！
>
>
>
> Best regards,
>
>
>
> Authors

---

> ### Author Response · Authors · 2024-12-02
> **Follow-Up on Your Review**
>
> Dear Reviewer cdNJ,
>
> Thank you once again for your thoughtful and constructive feedback, which has been instrumental in refining our work. Your initial comments were incredibly valuable, and we deeply appreciate the high evaluation you have already given our submission.
>
> As the discussion phase comes to a close, we wanted to kindly ask if you have any additional suggestions or feedback that could further improve our work. **Additionally, we would love to hear whether our responses and updates have satisfactorily addressed your concerns.**
>
> (*To briefly reiterate our paper's contribution, we identified a common issue in existing model editing methods—the disruption of stored correct knowledge—caused by overfitting to new knowledge, leading to distributional shifts in hidden representations. **Our proposed AlphaEdit addresses this issue comprehensively across current methods, achieving this with just a single line of code.***)
>
> We are sincerely grateful for the time and expertise you have shared, and we look forward to any final thoughts you might have.
>
>
> Best regards,
>
>
> The Authors

---

> > ### Comment · Reviewer_cdNJ · 2024-12-02
> > **Response to Authors**
> >
> > Thanks to the authors for the detailed response! The response addresses my concern with the correlation between dataset size and accuracy. I also like the updated figure 12 and it clearly illustrates the distribution.
> >
> > For figure 7a, I am still slightly unsure what does the 'the categories of knowledge ' in ylabel mean. For specifically, what does the words in y axis -- 'language', 'was born in', 'headquartered', 'produced by' etc mean?

---

> > > ### Author Response · Authors · 2024-12-03
> > > **Response to Your Latest Comment**
> > >
> > > Dear Reviewer cdNJ,
> > >
> > > Thank you so much for your prompt and thoughtful response! We are delighted that the additional experiments and updated figures align with your expectations and have addressed your concerns.
> > >
> > > Regarding your question about the y-axis labeled "categories of knowledge" in Figure 7a:
> > >
> > > - This axis represents the editing success rate of AlphaEdit for knowledge belonging to different semantic categories. For instance, the bar labeled "language" with a height of 98 indicates that out of 1,000 knowledge instances related to "language" (e.g., "*The primary language in the United States is English*," or "*The official language of France is French*"), AlphaEdit successfully edited 98% of them.
> > >
> > > This metric provides a fine-grained assessment of AlphaEdit's effectiveness across various knowledge domains, offering a clearer picture of its advantages.
> > >
> > > **In response to your comment, we have clarified the axes and experimental details for this figure in the revised manuscript.** Specifically, the updates can be found in `Section 4.4` (Lines 466–469) and `Appendix C.4` (Lines 1299–1304). However, since we are unable to upload a new version at this stage, we will ensure these clarifications are reflected in the camera-ready version.
> > >
> > > We sincerely hope this response could address your concern. Once again, thank you for your timely and insightful feedback—it has been invaluable in enhancing our work!
> > >
> > >
> > > Best regards,
> > >
> > >
> > > The Authors

---

> > > > ### Comment · Reviewer_cdNJ · 2024-12-03
> > > > **Final Response**
> > > >
> > > > Thank you authors for the clarification. Yes, please do add the clarification in the final camera-ready version since it was difficult to read without the explanation you described now. I do not have any other critical concerns. Good luck :)

---

> ### Author Response · Authors · 2024-12-03
> **Thank You for Your Feedback**
>
> Dear Reviewer cdNJ,
>
> Thank you for your kind response and for taking the time to review our clarification. We greatly appreciate your understanding and constructive feedback, which have been invaluable throughout the review process.
>
>
> Your thoughtful comments have significantly contributed to improving our work, and your encouragement means a great deal to us. We sincerely thank you for your support and guidance.
>
>
> Best regards,
>
> The Authors

---

### Official Review · Reviewer_JPxS · 2024-11-04

**Soundness:** 3
**Presentation:** 3
**Contribution:** 3
**Rating:** 8
**Confidence:** 3

**Summary:**

The paper introduces AlphaEdit, a method to improve targeted knowledge editing in large language models by projecting updates onto the null space of preserved knowledge, thus reducing interference with existing information. AlphaEdit achieves this with a minimal adjustment in code, enabling it to maintain a model’s pre-existing knowledge while updating targeted information. Experimental results demonstrate AlphaEdit's effectiveness, showing a performance improvement over traditional editing methods across multiple language models.

**Strengths:**

1. The use of null-space projection in AlphaEdit minimizes disruption to preserved knowledge while updating new information, effectively addressing a common trade-off in model editing between knowledge update and retention.
2. The paper provides comprehensive experimental evidence that AlphaEdit outperforms existing methods on critical editing metrics such as efficacy, generalization, specificity, fluency, and consistency.

**Weaknesses:**

1. Accurate null-space projection may rely on high-dimensional matrix computations, which could pose scalability issues as model sizes or knowledge bases grow.
2. Limited empirical evaluation on diverse LLMs. The method is tested on models like GPT-2 XL, GPT-J, and LLaMA3 only. It would be good to see results for other models such as gemma, phi.

**Questions:**

1. Authors may have overlooked these methods. There are other latest methods present such as SERAC, GRACE, InstructEdit, MELO methods. Authors either provide the compared results or argue on why they have not considered these methods for comparison.
https://sites.google.com/view/serac-editing
https://arxiv.org/abs/2211.11031
https://arxiv.org/abs/2402.16123
https://arxiv.org/abs/2312.11795

2. Can authors show the results on  KnowEdit dataset as well?

---

> ### Author Response · Authors · 2024-11-19
> **Response to Reviewer JPxS (1/4)**
>
> Dear Reviewer JPxS:
>
>
> Thank you for your positive feedback and valuable suggestions! We sincerely appreciate the time and effort you have dedicated to reviewing our work. Below, we meticulously provide responses to each of your comments and outline the modifications based on your suggestions. All revisions are highlighted in blue.
>
> ------------
>
>
> ## *W1: null-space projection may rely on high-dimensional matrix computations, which could pose scalability issues as model sizes or knowledge bases grow.*
>
>
>
> Thank you for raising this important concern! In fact, **the computational complexity of null-space projection in AlphaEdit is unaffected by the base LLM’s size and knowledge base**. We provide our reasoning from both theoretical analysis and experimental validation below:
>
>  1.  Theoretical Analysis:
>        - As stated in the original manuscript (Line 191–192), implementing null-space projection in AlphaEdit only requires calculating the null-space projection matrix for $K_0 K_0^T \in \mathbb{R}^{d_0 \times d_0}$. The computational complexity of this calculation **depends solely on the hidden dimension $d_0$   of the base LLM**, and is independent of the model size and the knowledge base. Furthermore, the hidden dimensions $d_0$ of commonly used LLMs are typically in the range of a few thousand, thus the time cost for null-space projection is negligible compared to the gradient descent time cost of baseline methods such as MEMIT.
>
>  2. Experimental Validation:
>        - To empirically verify our claims and analysis, we tested the average runtime for 100 edits performed by AlphaEdit on three different LLMs with varying model sizes and knowledge bases (LLaMA3, GPT-J, and GPT2-XL). The results are summarized below:
>
> ||||||||
> |:-:|:-:|:-:|:-:|:-:|:-:|:-:|
> |$\text{Method}$||$\text{Counterfact}$|||$\text{ZsRE}$||
> ||$\text{LLaMA3}$|$\text{GPT-J}$|$\text{GPT2-XL}$|$\text{LLaMA3}$|$\text{GPT-J}$|$\text{ GPT2-XL}$|
> |$\text{MEMIT}$|$222.51\text{s}$|$334.74\text{s}$|$474.14\text{s}$|$231.32\text{s}$|$344.21\text{s}$|$488.37\text{s}$|
> |$\text{AlphaEdit}$|$223.24\text{s}$|$336.93\text{s}$|$476.79\text{s}$|$231.40\text{s}$|$345.52\text{s}$|$490.25\text{s}$|
> ||||||||
>
>
>
>
> From the table, we observe that across LLMs of varying sizes and knowledge bases, **AlphaEdit does not incur additional time costs** compared to MEMIT. This validates the scalability of our null-space projection approach.
>
>
>
> Additionally, in response to your helpful comment, we have added a new subsection to the revised manuscript (`Appendix C.9`, `Lines 1404–1412`), where we present both the theoretical analysis and experimental results on AlphaEdit’s runtime. We hope this addition will address similar concerns from other readers.
>
>
>
> Hope our response could address your concern!

---

> ### Author Response · Authors · 2024-11-19
> **Response to Reviewer JPxS (2/4)**
>
> ## *W2:  It would be good to see results for other models such as gemma and phi.*
>
> Thank you for your valuable suggestion! Following your feedback, we have expanded our experiments to include two additional base LLMs, **Gemma** and **phi-1.5**, as you recommended. These new results are now included in the revised manuscript (`Appendix C.7`, Lines 1347–1401).
>
>
> For your convenience, we provide a summary of the representative results and analysis below:
>
>
>
> |||||||
> |:-:|:-:|:-:|:-:|:-:|:-:|
> ||||$\text{Counterfact}$|||
> |$\text{Method}$|$\text{Model}$|$\text{Eff.}\uparrow$|$\text{Gen.}\uparrow$|$\text{Spe.}\uparrow$|$\text{Flu.}\uparrow$|$\text{Consis.}\uparrow$|
> |$\text{MEMIT}$||$64.68\pm0.21$|$60.36\pm0.30$|$46.73\pm0.62$|$373.94\pm1.12$|$22.14\pm0.31$|
> |$\text{RECT}$|$\text{Gemma}$|$65.17\pm0.19$|$57.48\pm0.64$|$52.54\pm0.54$|$388.77\pm0.44$|$23.37\pm0.39$|
> |$\text{AlphaEdit}$||$\mathbf{75.21\pm0.09}$|$\mathbf{67.83\pm0.63}$|$\mathbf{52.63\pm0.49}$|$\mathbf{398.96\pm0.39}$|$\mathbf{32.91\pm0.35}$|
> |||||||
> |$\text{MEMIT}$||$55.71\pm1.63$|$56.58\pm0.78$|$35.41\pm0.99$|$368.57\pm1.26$|$19.79\pm0.31$|
> |$\text{RECT}$|$\text{phi-1.5}$|$58.19\pm0.73$|$58.92\pm0.76$|$38.46\pm0.92$|$362.94\pm1.44$|$19.88\pm0.37$|
> |$\text{AlphaEdit}$||$\mathbf{70.79\pm0.56}$|$\mathbf{65.12\pm0.88}$|$\mathbf{48.96\pm0.96}$|$\mathbf{399.47\pm0.67}$|$\mathbf{25.98\pm0.48}$|
> |||||||
>
>
> |||||||
> |:-:|:-:|:-:|:-:|:-:|:-:|
> |||$\text{ZsRE}$||||
> |$\text{Method}$|$\text{Model}$|$\text{Eff.}\uparrow$|$\text{Gen.}\uparrow$|$\text{Spe.}\uparrow$|
> |$\text{MEMIT}$||$64.38\pm0.26$|$66.12\pm0.46$|$\mathbf{24.52\pm0.38}$|
> |$\text{RECT}$|$\text{Gemma}$|$67.18\pm0.50$|$64.12\pm0.47$|$20.02\pm0.47$|
> |$\text{AlphaEdit}$||$\mathbf{75.91\pm0.42}$|$\mathbf{68.12\pm0.67}$|$23.50\pm0.56$|
> |||||||
> |$\text{MEMIT}$||$54.41\pm0.78$|$52.47\pm0.89$|$\mathbf{20.98\pm0.58}$|
> |$\text{RECT}$|$\text{phi-1.5}$|$55.15\pm0.72$|$53.64\pm0.83$|$18.58\pm0.65$|
> |$\text{AlphaEdit}$||$\mathbf{70.02\pm0.85}$|$\mathbf{63.19\pm0.72}$|$20.69\pm0.73$|
> |||||||
>
>
> According to the above Table, we can find that:
>
> - **AlphaEdit consistently outperforms MEMIT and RECT across key metrics** on both the Counterfact and ZsRE datasets. Notably, on Gemma, AlphaEdit achieves the highest fluency (398.96) and consistency (32.91), reflecting its ability to maintain coherence and accuracy. Similarly, on phi-1.5, AlphaEdit excels in efficacy (70.79) and fluency (399.47), showcasing its adaptability to smaller, efficient models.
>
> These findings demonstrate AlphaEdit’s generalizability across **a wider range of LLM architectures**, underscoring its ability to deliver high-quality edits while preserving model integrity.
>
> Hope our additional experiments could resolve your concern!
>
> ------------
>
>
>
> ## *Q1: Authors may have overlooked these methods: SERAC, GRACE, InstructEdit, and MELO when providing the comparison results.*
>
>
>
> Thank you for highlighting these important methods! We sincerely appreciate your effort to help us improve the manuscript.
>
> In the original submission, we did not include comparisons with methods such as SERAC, GRACE, and MELO, as they primarily focus on **parameter-preserving strategies that require additional memory modules**, whereas AlphaEdit is specifically designed as a parameter-modifying editing method that directly alters model parameters.
>
>
>
> That said, your insightful comment has helped us recognize the potential value of including such comparisons for a more comprehensive evaluation. In response to your valuable input:
>
>
>
>  1. We have provided detailed descriptions of SERAC, GRACE, InstructEdit, and MELO in `Related Work`  (Line 511–515) and `Experimental Setup` (Line 344-347, 810-815 and 834–847) of the revised manuscript;
>
>  2. We have employed SERAC, GRACE, InstructEdit, and MELO to conduct additional experiments, and presented the corresponding results and analysis in `Section 4.2` (Line 280, 289 and 299) and `Appendix C.5` (Line 1296–1319).

---

> ### Author Response · Authors · 2024-11-19
> **Response to Reviewer JPxS (3/4)**
>
> To provide a quick overview, we summarize some of the results and analysis below:
>
>
> |||||||||||
> |:-:|:-:|:-:|:-:|:-:|:-:|:-:|:-:|:-:|:-:|
> |$\text{Method}$|$\text{Model}$|||$\text{Counterfact}$||||$\text{ZsRE}$||
> |||$\text{Eff.}\uparrow$|$\text{Gen.}\uparrow$|$\text{Spe.}\uparrow$|$\text{Flu.}\uparrow$|$\text{Consis.}\uparrow$|$\text{Eff.}\uparrow$|$\text{Gen.}\uparrow$|$\text{Spe.}\uparrow$|
> ||||||||||||
> |$\text{Pre-edited}$|$\text{LLaMA3}$|$7.85\pm0.26$|$10.58\pm0.26$|$89.48\pm0.18$|$635.23\pm0.11$|$24.14\pm0.08$|$36.99\pm0.30$|$36.34\pm0.30$|$31.89\pm0.22$|
> |$\text{InstructEdit}$|$\text{LLaMA3}$|$66.58\pm0.24$|$64.18\pm0.35$|$47.14\pm0.37$|$443.85\pm0.78$|$7.28\pm0.04$|$1.58\pm0.04$|$1.36\pm0.08$|$1.01\pm0.05$|
> |$\text{SERAC}$|$\text{LLaMA3}$|$71.21\pm0.56$|$61.05\pm0.39$|$66.90\pm0.21$|$615.72\pm0.34$|$20.77\pm0.13$|$67.75\pm0.24$|$33.96\pm0.35$|$22.17\pm0.15$|
> |$\text{GRACE}$|$\text{LLaMA3}$|$96.72\pm0.13$|$50.14\pm0.01$|$\mathbf{72.23\pm0.21}$|$620.43\pm0.63$|$23.79\pm0.23$|$93.58\pm0.31$|$1.03\pm0.06$|$31.86\pm0.12$|
> |$\text{MELO}$|$\text{LLaMA3}$|$65.29\pm0.13$|$58.58\pm0.32$|$63.36\pm0.37$|$608.98\pm0.82$|$22.18\pm0.04$|$25.18\pm0.14$|$24.14\pm0.23$|$30.36\pm0.75$|
> |$\text{AlphaEdit}$|$\text{LLaMA3}$|$\mathbf{98.90\pm0.10}$|$\mathbf{94.22\pm0.19}$|$67.88\pm0.29$|$\mathbf{622.49\pm0.16}$|$\mathbf{32.40\pm0.11}$|$\mathbf{94.47\pm0.13}$|$\mathbf{91.13\pm0.19}$|$\mathbf{32.55\pm0.22}$|
> ||||||||||||
> |$\text{Pre-edited}$|$\text{GPT-J}$|$16.22\pm0.31$|$18.56\pm0.45$|$83.11\pm0.13$|$621.81\pm0.67$|$29.74\pm0.51$|$26.32\pm0.37$|$25.79\pm0.25$|$27.42\pm0.53$|
> |$\text{InstructEdit}$|$\text{GPT-J}$|$50.62\pm0.58$|$51.73\pm0.42$|$56.28\pm0.50$|$245.89\pm0.44$|$4.21\pm0.04$|$0.92\pm0.07$|$0.88\pm0.03$|$0.65\pm0.06$|
> |$\text{SERAC}$|$\text{GPT-J}$|$82.28\pm0.26$|$58.31\pm0.34$|$68.98\pm0.32$|$615.92\pm0.72$|$28.65\pm0.17$|$92.37\pm0.29$|$38.21\pm0.32$|$25.17\pm0.25$|
> |$\text{GRACE}$|$\text{GPT-J}$|$96.50\pm0.24$|$50.10\pm0.01$|$74.42\pm0.43$|$\mathbf{620.56\pm0.79}$|$31.55\pm0.25$|$96.54\pm0.21$|$0.40\pm0.02$|$24.78\pm0.21$|
> |$\text{MELO}$|$\text{GPT-J}$|$78.29\pm0.24$|$60.52\pm0.52$|$66.80\pm0.52$|$610.82\pm0.44$|$24.31\pm0.24$|$82.24\pm0.07$|$32.88\pm0.03$|$26.65\pm0.24$|
> |$\text{AlphaEdit}$|$\text{GPT-J}$|$\mathbf{99.75\pm0.08}$|$\mathbf{96.38\pm0.23}$|$\mathbf{75.48\pm0.21}$|$618.50\pm0.17$|$\mathbf{42.08\pm0.15}$|$\mathbf{99.79\pm0.14}$|$\mathbf{96.00\pm0.22}$|$\mathbf{28.29\pm0.25}$|
> ||||||||||||
> |$\text{Pre-edited}$|$\text{GPT2-XL}$|$22.23\pm0.73$|$24.34\pm0.62$|$78.53\pm0.32$|$626.64\pm0.31$|$31.88\pm0.20$|$22.19\pm0.24$|$31.30\pm0.27$|$24.15\pm0.32$|
> |$\text{InstructEdit}$|$\text{GPT2-XL}$|$55.32\pm0.58$|$53.63\pm0.42$|$53.25\pm0.62$|$412.57\pm0.15$|$1.08\pm0.03$|$3.54\pm0.03$|$4.25\pm0.02$|$3.23\pm0.04$|
> |$\text{SERAC}$|$\text{GPT2-XL}$|$72.25\pm0.15$|$58.18\pm0.32$|$64.06\pm0.37$|$595.35\pm0.35$|$27.35\pm0.12$|$92.17\pm0.67$|$36.57\pm0.72$|$20.67\pm0.22$|
> |$\text{GRACE}$|$\text{GPT2-XL}$|$98.88\pm0.13$|$50.05\pm0.01$|$\mathbf{72.07\pm0.24}$|$\mathbf{620.21\pm0.49}$|$28.53\pm0.15$|$94.33\pm0.37$|$1.59\pm0.03$|$\mathbf{27.63\pm0.43}$|
> |$\text{MELO}$|$\text{GPT2-XL}$|$72.62\pm0.58$|$53.63\pm0.42$|$63.25\pm0.36$|$588.57\pm0.65$|$23.58\pm0.33$|$93.54\pm0.03$|$45.25\pm0.02$|$23.45\pm0.24$|
> |$\text{AlphaEdit}$|$\text{GPT2-XL}$|$\mathbf{99.50\pm0.04}$|$\mathbf{93.95\pm0.34}$|$66.39\pm0.31$|$597.88\pm0.18$|$\mathbf{39.38\pm0.15}$|$\mathbf{94.81\pm0.30}$|$\mathbf{86.11\pm0.29}$|$25.88\pm0.21$|
> ||||||||||||
>
>
>
>
> From the above table, we observe the following key findings:
>
>
>  1.  Across all base LLMs and datasets, AlphaEdit consistently achieves the highest scores in efficacy (Eff.) and generalization (Gen.). This indicates that AlphaEdit is highly effective at correctly applying the desired edits while maintaining robust generalization to the relevant knowledge.
>
>  2. While AlphaEdit generally achieves competitive scores in specificity (Spe.) and fluency (Flu.), it does not always surpass the memory-based methods. However, we believe **this trade-off is reasonable and acceptable because memory-based methods inherently rely on consuming storage space to better preserve existing knowledge**.
>
>
>
> If there are any other models you would like us to discuss or include as baselines, we would be more than happy to conduct additional experiments to incorporate them!
>
>
> Hope our response could address your concerns!

---

> ### Author Response · Authors · 2024-11-19
> **Response to Reviewer JPxS (4/4)**
>
> ## *Q2: Can authors show the results on KnowEdit dataset as well?*
>
> Thanks for your valuable suggestion! Following you feedback:
>
>  1. We have provided detailed descriptions of **KnowEdit** in `Related Work` (Line 516–518) and `Experimental Setup` (Line 354–355 &  721–727) of the revised manuscript;
>
>  2. We have conducted additional experiments using two representative datasets from the KnowEdit database—**wiki_recent** and **wikibio**. The corresponding results and analyses have been included in `Appendix C.7` (Line 1347–1395).
>
> For your convenience, we summarize some of the results and analysis below:
>
> |||||||||
> |:-:|:-:|:-:|:-:|:-:|:-:|:-:|:-:|
> |$\text{Method}$|||$\text{wiki-recent}$|||$\text{wikibio}$|||
> ||$\text{Edit Succ.}\uparrow$|$\text{Portability}\uparrow$|$\text{Locality}\uparrow$|$\text{Fluency}\uparrow$|$\text{Edit Succ.}\uparrow$|$\text{Locality}\uparrow$|$\text{Fluency}\uparrow$|
> |$\text{MEMIT}$|$56.25\pm0.28$|$42.73\pm0.27$|$41.02\pm0.20$|$513.35\pm3.47$|$63.73\pm0.40$|$64.27\pm0.41$|$582.38\pm3.34$|
> |$\text{RECT}$|$82.47\pm0.53$|$51.28\pm0.25$|$48.84\pm0.24$|$568.62\pm3.71$|$91.48\pm0.48$|$72.83\pm0.44$|$612.04\pm4.29$|
> |$\text{AlphaEdit}$|$\mathbf{96.10\pm0.47}$|$\mathbf{57.30\pm0.38}$|$\mathbf{54.76\pm0.30}$|$\mathbf{594.52\pm3.91}$|$\mathbf{95.34\pm0.46}$|$\mathbf{75.34\pm0.50}$|$\mathbf{618.35\pm4.22}$|
> |||||||||
>
>
>
>
> According to the above Table, we can find that AlphaEdit demonstrates a remarkable ability to achieve high editing success rates across wiki_recent and wikibio. For instance, on the wiki_recent dataset, AlphaEdit achieves an impressive 96.10\% editing success, which is significantly higher than the second-best method, RECT (82.47\%).
>
>
> Additionally, to provide a more comprehensive evaluation of AlphaEdit’s performance, we have also introduced the following experiments in the revised manuscript:
>
>  1. Added **LEME** [1] dataset to assess the performance of AlphaEdit across different output text lengths.
>
>  2. Added **MQuAKE** [2] dataset to evaluate the ability of AlphaEdit to answer multi-hop knowledge-based questions.
>
>
> If you are interested, we warmly encourage you to refer to `Appendices C.7` for the complete results and detailed analysis.
>
>
>
> Hope that these updates could meet your expectations, and **we would be thrilled if you could let us know whether your concerns have been addressed or if you have any follow-up questions!**
>
>
> --------------------------
>
>
>
> Once again, we deeply appreciate your thoughtful and encouraging feedback. Your suggestions have not only enhanced the current work but have also inspired us to to **keep moving forward and contributing to the community**!
>
>
> Best,
>
>
> Authors of Paper 3792
>
>
>
>
>
>
>   *[1] Long-form evaluation of model editing. 2024*
>
>   *[2] MQuAKE: Assessing Knowledge Editing in Language Models via Multi-Hop Questions. 2023*

---

> ### Comment · Reviewer_JPxS · 2024-11-22
> **Comment by Reviewer JPxS**
>
> Thank you for incorporating the suggestions from the review. I have increased the rating to 8.

---

> > ### Author Response · Authors · 2024-11-23
> >
> > Dear Reviewer JPxS,
> >
> >
> > Thank you for your kind feedback and for taking the time to review our updated work. We are grateful for your recognition and for increasing the rating—it means a lot to us and inspires us to continue improving.
> >
> >
> > **We are deeply committed to advancing the field of efficient knowledge updates for LLMs. Your valuable comments have  helped us refine our work, and we are excited to keep contributing meaningful insights and solutions to this area.**
> >
> >
> > Thank you again for your thoughtful comments and encouragement. We genuinely appreciate your support.
> >
> >
> > Best regards,
> >
> >
> > Authors

---

### Official Review · Reviewer_ngVy · 2024-11-04

**Soundness:** 3
**Presentation:** 3
**Contribution:** 3
**Rating:** 8
**Confidence:** 3

**Summary:**

This is a review of the paper entitled “AlphaEdit: Null-Space Constrained Knowledge Editing for Language Models” submitted to ICLR 2025. The paper suggests a new approach to do targeted knowledge updates in LLMs; in particular, if an LLM tells some wrong factual information, the goal is to identify influential parameters and then introduce so-called perturbation to them that, on the one hand, repairs problematic outputs and, on the other, keeps the rest as intact as possible. The main experimental result is that the new suggested method, called AlphaEdit, performs comparably to the state of the art for single updates and shorts sequences of updates, but outperforms them dramatically for longer ones.

**Strengths:**

I should start by admitting that I am not a specialist in the topic of the paper, and so it is difficult for me to judge the novelty and value of the results. However, I can say that the paper is well-written: I could understand nearly everything and agree with the arguments. Moreover, for an outsider, the results look interesting and promising. Thus, I lean towards acceptance; however, of course, the opinions of reviewers who are more in the topic should be more valuable for the decision.

**Weaknesses:**

—

**Questions:**

Concrete comments:

Question: why do we need to solve the sequential editing task really sequentially, as in L (line) 244—that is, why cannot we just start from scratch with the original model and K1 being all the edits together (i.e., the union of Kp and K1 in the current equation (12)), and use Equation (11)?
This seems to promise a better performance even for the AlphaEdit, looking at Figure 4.

Minor:
L 11: I do not think “due” is the right word here.
L 76: “the coefficient” is unclear. Which coefficient?
L 81: besides viewing in colour, one needs a magnifier to read these figures.
L 136: “update” and “new” do not make much sense together, either one or another.
L 173: B -> A
L 177 (equation (7)): \Delta is not really \Delta here, it is projected \Delta. Which should be said or better denoted by another symbol.
L 193: it is not clear what does it mean to be “consistent” in this context

---

> ### Author Response · Authors · 2024-11-19
> **Response to Reviewer ngVy**
>
> Dear Reviewer ngVy:
>
>
> Thank you for your kind words and positive feedback! Your approval is the great encouragement for us and motivates us to continue advancing our work.
>
>
> Below, we meticulously provide responses to each of your comments and outline the modifications made to the manuscript. All revisions are highlighted in blue.
>
>
>
>
> -----------
>
>
>
>
>
>
> ## *Q1. Why do we need to solve the sequential editing task, that is, why cannot we just start from scratch and K1 being all the edits together?*
>
> Great catch! Your question highlights a fundamental issue in model editing: **why we need sequential editing rather than starting from scratch and conducting all the edits at once?** Upon reviewing the current literature, we summarize two main reasons for this:
>
>
> 1. Time and Computational Efficiency: In real-world scenarios, new knowledge      constantly emerges. Sequential editing allows us to incorporate only the      new knowledge each time, resulting in a computational cost of 1 + 1 + 1 + ... + 1 (N times) for N updates. In contrast, starting from scratch requires re-editing all previous knowledge with each new addition, leading  to a total cost of 1 + 2 + 3 + ... + N, which becomes unsustainable as N grows.
>
>
> 2. Privacy Concerns: In certain scenarios where privacy and security are critical, model users may not want previously injected knowledge to be visible to future users.      Sequential editing can address this need perfectly. In contrast, starting from scratch would require accessing all previously edited knowledge with each new addition, potentially compromising privacy.
>
> Beyond these points, sequential editing also requires less memory than starting from scratch, as it avoids the need to store all prior knowledge.
>
> In a nutshell, **sequential editing is far more efficient in terms of time, computation, and memory, and it can be applied to a wider range of real-world scenarios**. This is why research into sequential editing has gained increasing prominence more recently.
>
>
> However, it's worth noting that while sequential editing has many advantages, earlier approaches faced a key challenge: as the number of edits grows, cumulative changes can degrade the model, potentially reducing its performance. Some prior methods, such as PRUNE, made attempts to mitigate this issue, but the problem persists with increasing edits. Our approach, AlphaEdit, **addresses this issue with a minimal adjustment in code**, which we believe is the main contribution of our paper to the community.
>
> Hope our response could address your concerns!
>
>
>
> --------
>
>
>
>
>
> ## *Q2. Minor: (1) “due” in L 11 is not right. (2) “coefficient” in L 76 is unclear. (3) A magnifier is needed to readfigures in L 81. (4) “update” & “new” in L 136, either one or another. (5) L 173: replace B with A. (6) L 177: the projected $\Delta$ should be said by another symbol. (7) L 193: What “consistent” means here is unclear.*
>
> Thank you for your suggestions! Based on your suggestions, we have made the following revisions in the updated manuscript:
>
>  1. L11：Replaced "due to" with "producing": LLMs often exhibit hallucinations, producing incorrect or outdated knowledge.
>
>  2.  L76：Added clarification for "coefficient": $\lambda$ is the coefficient to keep balance between $e_0$ and $e_1$ in the objective.
>
>  3. L81：Added reference to detailed numerical results: Detailed settings and results are provided in Section 4.2 and Table 1, respectively.
>
>  4. L 136: Removed "new": each edit needs to update u pieces of knowledge in the form of (s, r, o).
>
>  5. L 173：Replaced "B" with the correct symbol "A": B is in the null space of A if and only if BA=0.
>
>  6. L 177：Introduced symbol $\Delta’$ to represent the projected \Delta：$\Delta'K_0=0$, where $\Delta’$ denotes the projected perturbation.
>
>  7. L 193: Replaced "consistent" with "equal to": This matrix's null space is equal to that of $K_0$.
>
> Hope that these updates could meet your expectations, and **we would be thrilled if you could let us know whether your concerns have been addressed or if you have any follow-up questions!**
>
> --------------------------
>
>
> Once again, we deeply appreciate your thoughtful and encouraging feedback. Your suggestions have not only enhanced the current work but have also inspired us to to **keep moving forward and contributing to the community**!
>
>
> Best,
>
>
> Authors of Paper 3792

---

> ### Author Response · Authors · 2024-11-25
> **Heartfelt Gratitude to Reviewer ngVy**
>
> Dear Reviewer ngVy,
>
> We greatly appreciate for your positive feedback and constructive suggestions, which have been instrumental in improving the quality of our work.
>
>
> If you have any additional questions or concerns that we can clarify or address, we would be happy to provide further information to ensure all aspects of our work are clear.
>
>
> Thank you once again for your valuable time and effort in reviewing our submission!
>
>
> Best regards,
>
>
> Authors

---

> ### Author Response · Authors · 2024-12-02
> **Follow-Up on Your Review**
>
> Dear Reviewer ngVy,
>
> Thank you once again for your thoughtful and constructive feedback, which has been instrumental in refining our work. Your initial comments were incredibly valuable, and we deeply appreciate the high evaluation you have already given our submission.
>
> As the discussion phase comes to a close, we wanted to kindly ask if you have any additional suggestions or feedback that could further improve our work. **Additionally, we would love to hear whether our responses and updates have satisfactorily addressed your concerns.**
>
> (*To briefly reiterate our paper's contribution, we identified a common issue in existing model editing methods—the disruption of stored correct knowledge—caused by overfitting to new knowledge, leading to distributional shifts in hidden representations. **Our proposed AlphaEdit addresses this issue comprehensively across current methods, achieving this with just a single line of code.***)
>
> We are sincerely grateful for the time and expertise you have shared, and we look forward to any final thoughts you might have.
>
>
> Best regards,
>
>
> The Authors

---

### Official Review · Reviewer_u8aL · 2024-11-06

**Soundness:** 4
**Presentation:** 3
**Contribution:** 4
**Rating:** 8
**Confidence:** 3

**Summary:**

This paper introduces AlphaEdit, a novel method for knowledge editing in large language models (LLMs). The primary goal of AlphaEdit is to enable targeted knowledge updates while minimizing the disruption of existing knowledge. The authors propose projecting perturbations onto the null space of the preserved knowledge before applying them to the model parameters. This approach theoretically ensures that the output of the edited LLM remains unchanged when queried about the preserved knowledge, thereby mitigating the issue of knowledge disruption. Extensive experiments on various LLMs, including LLaMA3, GPT2-XL, and GPT-J, demonstrate that AlphaEdit significantly boosts the performance of existing model editing methods by an average of 36.4% with minimal additional code.

**Strengths:**

- The concept of projecting perturbations onto the null space of preserved knowledge is innovative and addresses a significant challenge in the field of knowledge editing for LLMs. The theoretical foundation provided in the paper is robust and well-explained.
- The authors conduct extensive experiments on multiple representative LLMs, demonstrating the effectiveness of AlphaEdit. The performance improvements are substantial and consistent across different models.

**Weaknesses:**

1. Well, actually I think the work is great and I donot see any weakness, the thing is that I think the author can do more benchmarks like the LongformEvaluation, MQUAKE which consider some more knowledge utilization ablity for knowledge editing.
But the current evaluation is good enough.

[1] Long-form evaluation of model editing

[2] MQuAKE: Assessing Knowledge Editing in Language Models via Multi-Hop Questions

**Questions:**

N/A

---

> ### Author Response · Authors · 2024-11-19
> **Response to Reviewer u8aL**
>
> Dear Reviewer u8aL:
>
> Thank you for your kind words and positive feedback regarding the novelty, presentation, and effectiveness of our work! Your approval is the great encouragement for us and motivates us to continue advancing our research.
>
>
> We also appreciate your insightful suggestion to include additional benchmarks: ***LongformEvaluation*** and ***MQUAKE***. In response to your valuable input:
>
> - In the revised manuscript, we have provided detailed descriptions of these datasets in the `Related Work` (Line 516–521) and the `Experimental Setup` (Line 352–355 & 727–736).
>
> - We have conducted additional experiments on these two datasets, and presented the corresponding results and analysis in `Appendix C.7` (Line 1347–1401).
>
> All changes are highlighted in blue. For your convenience, we summarize some of the results and analysis below:
>
> ||||||||
> |:-:|:-:|:-:|:-:|:-:|:-:|:-:|
> ||||$\text{MQuAKE}$||$\text{Longform}$||
> |$\text{Model}$|$\text{Method}$|$\text{Multi-hop}$$\uparrow$|$\text{Multi-hop (CoT)} $$\uparrow$|$\text{Edit}$$\uparrow$|$\text{Factual}$$\uparrow$|$\text{Internal}$$\uparrow$|
> | |$\text{MEMIT}$|$3.35\pm0.07$|$6.13\pm0.12$|$2.11\pm0.18$|$2.02\pm0.17$|$3.84\pm0.29$|
> | $\text{GPT-J}$ |$\text{RECT}$|$3.77\pm0.04$|$7.61\pm0.20$|$2.24\pm0.20$|$2.62\pm0.19$|$4.07\pm0.31$|
> ||$\text{AlphaEdit}$|$\mathbf{5.03\pm0.16}$|$\mathbf{9.14\pm0.21}$|$\mathbf{3.34\pm0.26}$|$\mathbf{3.80\pm0.28}$|$\mathbf{5.42\pm0.41}$|
> ||||||||
> | |$\text{MEMIT}$|$3.14\pm0.08$|$6.25\pm0.11$|$1.92\pm0.22$|$2.31\pm0.20$|$3.85\pm0.34$|
> | $\text{GPT2-XL}$ |$\text{RECT}$|$3.72\pm0.06$|$7.48\pm0.24$|$2.12\pm0.26$|$2.60\pm0.21$|$4.13\pm0.29$|
> ||$\text{AlphaEdit}$|$\mathbf{5.00\pm0.23}$|$\mathbf{9.25\pm0.27}$|$\mathbf{3.28\pm0.36}$|$\mathbf{3.07\pm0.33}$|$\mathbf{5.76\pm0.49}$|
>
>
>
> From the table, we observe the following key findings:
>
> 1. **On MQUAKE dataset, AlphaEdit demonstrates superior performance in both two metrics**, achieving scores of 9.14 and 9.75, respectively. These results significantly surpass competing methods, showcasing AlphaEdit’s capability in handling complex reasoning tasks while maintaining logical consistency across interdependent facts.
>
> 2. **On LongformEvaluation dataset, AlphaEdit excels in all three metrics**, reflecting its ability to generate accurate long-form outputs. Its consistently high performance across GPT-J and GPT2-XL highlights its reliability in executing precise edits while preserving the structural coherence and integrity of the generated text.
>
> Additionally, to provide a more comprehensive evaluation of AlphaEdit’s performance, we have also introduced the following experiments in the revised manuscript:
>
>   - New Datasets: Added wiki_recent and wikibio from KnowEdit database to evaluate performance across diverse knowledge      content types.
>
> - New Baselines: Incorporated four      new baselines—SERAC, GRACE, InstructEdit, and MELO,      spanning two distinct categories (hypernetwork-based and memory-based      approaches).
>
> - New Base LLMs: Integrated two      additional base LLMs—Gemma and Phi.
>
> - Runtime Evaluation: Conducted new      runtime assessments to measure the computational efficiency of AlphaEdit      across various base LLMs.
>
> If you are interested, we warmly encourage you to refer to `Appendices C.5, C.6, C.7, C.8`, and `C.9` for the complete results and detailed analysis.
>
>
> Hope that these updates could meet your expectations, and **we would be thrilled if you could let us know whether your concerns have been addressed or if you have any follow-up questions!**
>
>
> --------------------------
>
>
>
> Once again, we deeply appreciate your thoughtful and encouraging feedback. Your suggestions have not only enhanced the current work but have also inspired us to to **keep moving forward and contributing to the community**!
>
>
> Best,
>
>
> Authors of Paper 3792

---

> ### Author Response · Authors · 2024-11-25
> **Heartfelt Gratitude to Reviewer u8aL**
>
> Dear Reviewer u8aL,
>
> Thank you for your positive feedback and thoughtful suggestion to include the *Longform Evaluation* and *MQuAKE* datasets. We hope our updates could align with your expectations.
>
> **If you have any further questions or suggestions for improving our paper, we would be truly grateful to hear them.**
>
> Once again, we deeply appreciate the time and expertise you have dedicated to reviewing our work!
>
> Best regards,
>
> Authors of Paper 3792

---

> ### Author Response · Authors · 2024-12-02
> **Follow-Up on Your Review**
>
> Dear Reviewer u8aL,
>
> Thank you once again for your thoughtful and constructive feedback, which has been instrumental in refining our work. Your initial comments were incredibly valuable, and we deeply appreciate the high evaluation you have already given our submission.
>
> As the discussion phase comes to a close, we wanted to kindly ask if you have any additional suggestions or feedback that could further improve our work. **Additionally, we would love to hear whether our responses and updates have satisfactorily addressed your concerns.**
>
> (*To briefly reiterate our paper's contribution, we identified a common issue in existing model editing methods—the disruption of stored correct knowledge—caused by overfitting to new knowledge, leading to distributional shifts in hidden representations. **Our proposed AlphaEdit addresses this issue comprehensively across current methods, achieving this with just a single line of code.***)
>
> We are sincerely grateful for the time and expertise you have shared, and we look forward to any final thoughts you might have.
>
> Best regards,
>
> The Authors

---

### Author Response · Authors · 2024-11-19
**General Response**

Dear Reviewers,

We sincerely appreciate your time, efforts, and insightful feedback on our work! We are delighted that all reviewers recognized the motivation, novelty, presentation, and experimental effectiveness of our study.

In particular, we are grateful for your positive remarks about the significance of our work, such as `Reviewer u8aL`’s comment: "**AlphaEdit addresses a significant challenge in the field of knowledge editing**" and `Reviewer JPxS`’s note: "**AlphaEdit effectively addresses a common trade-off in model editing.**"

Below, we provide point-by-point responses to your comments and outline the revisions made to the manuscript based on your suggestions. All revisions are highlighted in blue. Notably, most comments suggest conducting additional experiments. In response, we have conducted a comprehensive set of new experiments, which we summarize here:

- *Three new datasets: KnowEdit, LEME (Longform Evaluation), and MQuAKE.*

- *Four new baselines: InstructEdit, SERAC, GRACE, and MELO.*

- *Two new base LLMs: Gemma and phi-1.5.*

- *Impact of dataset size: Relationship between the amount of data and AlphaEdit's performance.*

- *Runtime experiments: Execution time of AlphaEdit and baselines.*

We warmly encourage you to review the results in the revised manuscript. Hope our response and additional experiments could address your concerns!

Furthermore, please allow us to reiterate the key contribution of our work: **AlphaEdit addresses the common trade-off  in model editing—updating incorrect knowledge while preserving correct knowledge—with a single line of code**. We believe this contribution is crucial for advancing the field of editing LLMs, and we are truly grateful for your recognition of its significance.

Once again, we deeply appreciate the time and expertise you have shared with us. Your encouraging feedback  motivates us to continue advancing this work for the broader community, and we are more than happy to add clarifications to address any additional recommendations and reviews from you！

Best regards,

Authors of Paper 3792

---

### Author Response · Authors · 2024-12-04
**Appreciation and Reflections on the Discussion Phase**

Dear Area Chair and Reviewers,

Thank you for your support throughout the discussion phase! We deeply appreciate the time and effort you have dedicated to reviewing and discussing our submission.

**As the discussion phase comes to an end, we would like to take this opportunity to summarize the efforts we have made during this period.** In response to your suggestions, we conducted extensive additional experiments, including three new datasets, two additional base LLMs, and four new baselines.
We are pleased that these experiments have further validated our central claim: a common issue in existing model editing methods—the disruption of stored correct knowledge—is caused by overfitting to new knowledge, leading to distributional shifts in hidden representations. **Our method, AlphaEdit, addresses this challenge comprehensively across various  scenarios with just a single line of code, demonstrating its significance for advancing the model editing community.**

Once again, we sincerely appreciate the constructive discussions and the opportunity to refine our work. **Your contributions have been invaluable, and we look forward to continued engagement with the community!**


Best regards,


Authors of Paper 3792

---

### Public Comment · ~Weisen_Jiang1 · 2025-02-19
**question about the null space of $\mathbf{K}_0$**

Dear authors,

Congrats and thanks for this great work.

I have a question about the null space of $\mathbf{K}_0$.
As $\mathbf{K}_0$ is a $d_o\times 100,000$ matrix and $d_o<100,000$  (a flat matrix).
When $\mathbf{K}_0$ is full row rank (i.e., rank=$d_o$), its left null space is only the zero vector $\mathbf{0}$.
In this case, any update $\mathbf{\Delta}$ projected to the left null space of $\mathbf{K}_0$ will become $\mathbf{0}$.

So do we need to assume $\mathbf{K}_0$ has a rank smaller than $d_o$? if yes, why this assumption is reasonable?

Best,

---

> ### Public Comment · ~Junfeng_Fang1 · 2025-02-22
> **Clarification on Approximate Null Space in AlphaEdit**
>
> Dear Weisen Jiang,
>
> Thank you for your insightful question. Your observation regarding the null space of matrix $\mathbf{K}_0$ is indeed highly pertinent!
>
> In our work, we acknowledge that $\mathbf{K}_0$ is typically full-rank in most practical scenarios. As explicitly noted in the paper (see footnote on page 4), the null space we obtain is an approximate one. This approximation stems from the pronounced long-tail characteristic observed in the singular value distribution of $\mathbf{K}_0$. To address this, footnote on page 4 exhibits a threshold-based approach for singular value selection. The null space projection matrix is constructed by considering singular values below this predetermined threshold. This methodology ensures that the projected delta has a relatively minor impact on $\mathbf{K}_0$. However, it's crucial to note that the threshold selection requires careful consideration - while an excessively low threshold would unduly restrict the magnitude of parameter updates, too high a threshold might compromise the effectiveness of our approach.
>
> For practical implementation, we've provided empirical values in our GitHub repository (https://github.com/jianghoucheng/AlphaEdit) for reference. Additionally, our method has been integrated into the EasyEdit platform (https://github.com/zjunlp/EasyEdit), which you might find useful for experimentation and application.
>
> Please let me know if you have any further questions or need additional clarification.
>
> Best regards,
>
> Authors of Paper 3792

---

### Public Comment · ~Songshi_Liang1 · 2025-05-09
**Some questions about the projection**

I read your article carefully and found the research content very profound and valuable. However, I have a question. You said that we need to project the matrix onto the null space of k0. But will the appearance of projection matrix P affect the update effect of K1?

---

> ### Public Comment · ~Junfeng_Fang1 · 2025-05-11
>
> Thank you for your thoughtful question! You raise a critical theoretical concern. While our current empirical observations suggest that the projection matrix P does not significantly hinder the update effectiveness of K₁ in practice, this is largely attributed to the high-dimensional over-parameterization of LLMs, which inherently provides redundant degrees of freedom for parameter adjustments. However, we fully acknowledge that this assumption may not hold universally — for instance, in scenarios with low-rank knowledge updates or highly constrained parameter budgets, the impact of P could become non-negligible.
>
> This is indeed a vital direction we overlooked, and we strongly agree that it warrants deeper theoretical analysis (e.g., quantifying the rank reduction induced by P or exploring adaptive projection thresholds). Should you decide to delve into this direction, we would be happy to support your efforts.
>
> Thank you again for your sharp critique — it genuinely pushes the work forward.

---

### Meta-Review · Area_Chair_XH5b · 2024-12-20

**Metareview:**

This paper presents AlphaEdit, a novel method for knowledge editing in large language models (LLMs). The goal of AlphaEdit is to enable targeted updates to knowledge while minimizing disruption to existing information. The method involves projecting perturbations onto the null space of preserved knowledge before applying them to the model parameters, ensuring that the edited LLM’s output remains unchanged for queries related to preserved knowledge. Extensive experiments on various LLMs, including LLaMA3, GPT2-XL, and GPT-J, show that AlphaEdit improves the performance of existing editing methods by an average of 36.4% with minimal additional code.

This paper makes a significant contribution to the field of knowledge editing, presenting an elegant and well-designed method from a theoretical perspective. The proposed editing approach significantly reduces the impact on general capabilities, and the code is concise and elegant. All reviewers agree that the paper should be accepted for publication.

**Additional Comments On Reviewer Discussion:**

The authors provided many additional experiments during the rebuttal phase, and all reviewers gave the paper very high praise.

---

### Decision · Program_Chairs · 2025-01-22

Accept (Oral)